# A plug-and-play platform of ratiometric bioluminescent sensors for homogeneous immunoassays

Yan Ni [1,2,7], Bas J. H. M. Rosier [1,2,7], Eva A. van Aalen [1,2,7], Eva T. L. Hanckmann[1,2], Lieuwe Biewenga[1,2], Anna-Maria Makri Pistikou [2,3], Bart Timmermans[1,2], Chris Vu[1,2], Sophie Roos[1,2], Remco Arts[1,2], Wentao Li[4], Tom F. A. de Greef [1,2,3,5], Marcel M. G. J. van Borren[6], Frank J. M. van Kuppeveld [4], Berend-Jan Bosch [4] & Maarten Merkx [1,2 ✉]

Heterogeneous immunoassays such as ELISA have become indispensable in modern bioanalysis, yet translation into point-of-care assays is hindered by their dependence on external calibration and multiple washing and incubation steps. Here, we introduce RAPPID (Ratiometric Plug-and-Play Immunodiagnostics), a mix-and-measure homogeneous immunoassay platform that combines highly specific antibody-based detection with a ratiometric bioluminescent readout. The concept entails analyte-induced complementation of split NanoLuc luciferase fragments, photoconjugated to an antibody sandwich pair via protein G adapters. Introduction of a calibrator luciferase provides a robust ratiometric signal that allows direct in-sample calibration and quantitative measurements in complex media such as blood plasma. We developed RAPPID sensors that allow low-picomolar detection of several protein biomarkers, anti-drug antibodies, therapeutic antibodies, and both SARS-CoV-2 spike protein and anti-SARS-CoV-2 antibodies. With its easy-to-implement standardized workflow, RAPPID provides an attractive, fast, and low-cost alternative to traditional immunoassays, in an academic setting, in clinical laboratories, and for point-of-care applications.

[1] Laboratory of Chemical Biology, Department of Biomedical Engineering, Eindhoven University of Technology, Eindhoven, the Netherlands. [2] Institute for Complex Molecular Systems, Eindhoven University of Technology, Eindhoven, the Netherlands. [3] Computational Biology Group, Department of Biomedical Engineering, Eindhoven University of Technology, Eindhoven, the Netherlands. [4] Virology Section, Infectious Diseases and Immunology Division, Department of Biomolecular Health Sciences, Faculty of Veterinary Medicine, Utrecht University, Utrecht, the Netherlands. [5] Institute for Molecules and Materials, Radboud University, Nijmegen, the Netherlands. [6] Department of Clinical Chemistry, Rijnstate Hospital, Arnhem, the Netherlands. [7] These authors contributed equally: Yan Ni, Bas J.H.M. Rosier, Eva A. van Aalen. ✉email: m.merkx@tue.nl

The widely recognized need for more personalized, patient-centered healthcare has urged the development of cheap, easy-to-use, yet reliable biomolecular diagnostics. Such technology would not only accelerate biomolecular research in academic and clinical environments but more importantly, allow for fast diagnostic decision-making when access to clinical laboratories is unavailable or cost-prohibitive. Affinity-based immunoassays, such as ELISA and other heterogeneous immunoassays, form the cornerstone of today's clinical bioanalytical toolbox and offer both the specificity required to minimize crosstalk and the modularity to allow the application to a broad range of clinically relevant analytes. Translation of these immunoassays into point-of-care applications has proven challenging, due to the requirement of multiple washing and incubation steps, external calibration, and time-consuming optimization of sensor surface immobilization and assay conditions[1]. Lateral flow immunoassays (LFIAs) have been introduced as cheap and relatively easy-to-use point-of-care tests, but they suffer from limited sensitivity and are primarily used as qualitative tests[2,3].

Single-step affinity-based detection assays performed directly in solution could not only speed up traditional laboratory immunoassays but would be particularly attractive for point-of-care testing outside of the laboratory setting by non-expert users[4,5]. Several promising optical biosensor platforms have been reported that translate analyte binding directly into a conformational change, resulting in a change of signal intensity or color[6,7]. Of these, sensors based on bioluminescence are particularly suited for measurements directly in complex media such as blood with minimal sample handling, as they do not require external excitation and therefore do not suffer from issues related to autofluorescence or light scattering[8,9]. The Johnsson lab pioneered the development of semisynthetic sensor proteins (LUCID) based on bioluminescence resonance energy transfer (BRET) for the detection of small-molecule drugs such as methotrexate and digoxin, and important metabolites such as NAD[+] and NADPH[10–13]. Alternatively, the LUMABS platform was developed to enable the detection of specific antibodies directly in blood plasma with the use of a smartphone camera[14–17]. Both sensor platforms employ ratiometric bioluminescent detection, which allows quantitative measurements and enables integration in paper- and thread-based analytical devices for low-cost point-of-care detection[11,18,19]. However, successful development of LUCID and LUMABS sensors requires the availability of suitable ligand-binding domains and ligand analogs, or specific epitope peptides or mini-protein domains[20,21], respectively. Moreover, the change in emission ratio for these sensors is typically two to fourfold[14,17], and extensive protein engineering is required to obtain sensors with a larger dynamic range[13].

Here, we present RAPPID (ratiometric plug-and-play immunodiagnostics) as a comprehensive platform of ratiometric bioluminescent sensors that can be readily adapted to suit a very broad range of biomolecular targets. RAPPID is based on the complementation of antibody-conjugated split NanoLuc luciferases[22–25] to detect the formation of a sandwich immunocomplex in solution with a high intrinsic signal-to-background output (Fig. 1a). RAPPID does not require additional protein engineering and uses protein G-mediated photoconjugation to generate well-defined antibody conjugates directly from commercially available antibodies[26,27]. In addition, we introduce a green light-emitting calibrator luciferase as a simple method to provide a robust blue-over-green ratiometric readout and direct internal calibration. The ratiometric signal of RAPPID sensors is stable over time and less sensitive to experimental conditions such as temperature and substrate concentration, effectively addressing this fundamental problem of intensiometric bioluminescent assays. The ratiometric luminescent signal can be recorded in a one-step assay in real-time with a standard digital camera, highlighting the potential for point-of-care testing.

Development and implementation of a typical RAPPID assay for a new biomolecular target is straightforward and consists of three steps: (1) the selection of a pair of (commercial) antibodies that bind the target analyte, (2) crosslinking of the antibodies to the protein G-luciferase adapters using a simple one-hour illumination protocol, and (3) addition of both antibody-luciferase conjugates, the calibrator luciferase and the NanoLuc substrate to the sample, followed by detection of the emission ratio of blue over green light (Fig. 1a). In the case of antibody detection, the dynamic range of the assay and its scope can be further increased with an evolved set of RAPPID variants that use genetic fusions of the split luciferase components to the antibody's antigen. The broad scope and excellent analytical performance of RAPPID are demonstrated by developing assays for a range of clinically relevant biomarkers, including cardiac troponin I, C-reactive protein (CRP), three anti-drug-antibodies (ADAs), and two therapeutic antibodies, displaying robust increases in emission ratio of up to 36-fold. RAPPID detection of CRP was validated using a commercially available ELISA kit and in a clinical laboratory using 1-µL patient plasma samples. Finally, we demonstrate that the modularity of the RAPPID workflow enables prompt development of immunoassays for emerging target analytes such as SARS-CoV-2 spike proteins and anti-SARS-CoV-2 antibodies.

## Results

**Sensor design and characterization**. To establish a universally applicable sensing platform, we employed protein G-based photoconjugation for the synthesis of sensor components. This enables direct conjugation to most commercially available primary antibodies, obviating the use of labeled secondary antibodies which increases complexity and can decrease the efficiency of sandwich complex formation[24]. Both the 18-kDa large BiT (LB) and the 1.3-kDa small BiT (SB) fragment of split NanoLuc were genetically fused via a semi-rigid peptide linker to a protein G adapter domain (Gx), carrying the photocrosslinkable non-natural amino acid p-benzoylphenylalanine (Gx-LB and Gx-SB, respectively, see Fig. 1b and Supplementary Fig. 5). A variant of small BiT with $K_d = 2.5\,\mu M$ was chosen to ensure effective complementation in the sandwich complex (vide infra)[23]. Expression was performed in *Escherichia coli* with amber codon suppression, using co-transformation with a plasmid containing an engineered orthogonal aminoacyl tRNA synthetase/tRNA pair (available from AddGene, for a step-by-step protocol see Supplementary Information)[28,29]. Gx-LB and Gx-SB were obtained in yields similar to standard protein expression after affinity chromatography purification (respectively ~18 mg/L and ~5 mg/L, see Supplementary Fig. 6).

Next, antibody conjugation was performed by illuminating reaction mixtures with long-wavelength UV light ($\lambda = 365$ nm) in phosphate-buffered saline (pH 7.4) for 1 h on ice, covalently crosslinking Gx-LB and Gx-SB to the Fc-domain of the antibody. SDS-PAGE analysis revealed a mixture of mono-conjugated, bi-conjugated, and non-conjugated antibodies (Fig. 1b). Although the conjugation efficiency can be further improved by increasing the relative concentration of Gx-LB and Gx-SB, we typically chose to use an equal molar ratio of protein to antibody to minimize the presence of non-conjugated Gx-LB and Gx-SB, which could contribute to non-specific background complementation. If necessary, non-conjugated antibodies can be removed by small-scale $Ni^{2+}$-affinity chromatography, targeting the N-terminal polyhistidine tag of Gx-LB and Gx-SB (Fig. 1b). Since protein G binds to the majority of IgG-type antibodies, including all human

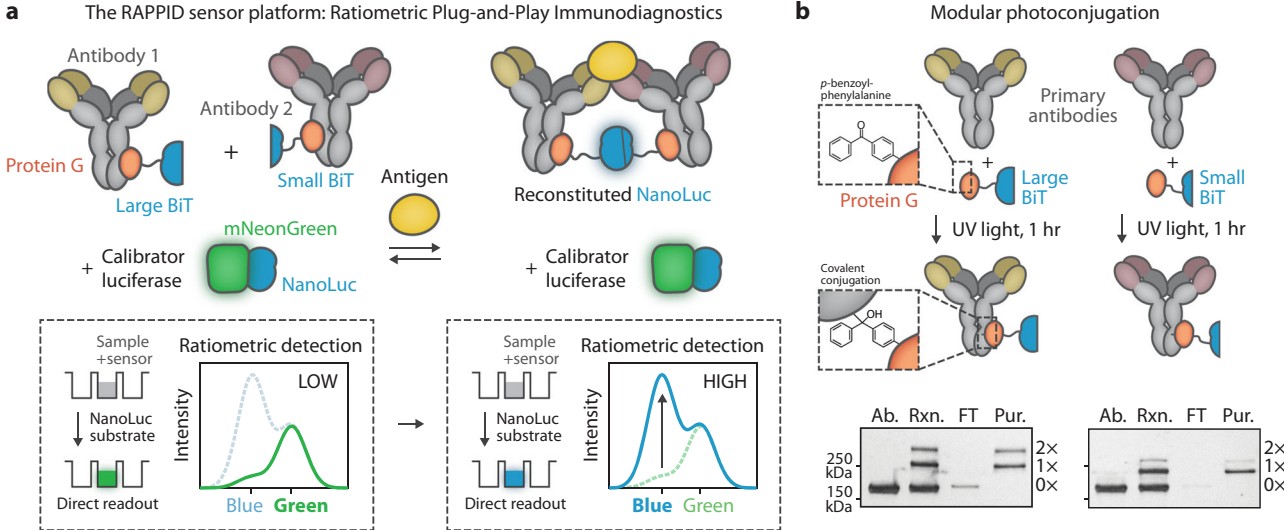

**Fig. 1 General overview and design elements of the RAPPID sensor platform. a** Schematic representation of the concept of antigen detection using a RAPPID sensor. A pair of primary antibodies is functionalized with a split version of the NanoLuc luciferase (large BiT and small BiT, respectively) through the use of protein G-based photoconjugation. Antigen binding induces colocalization of the antibody pair and subsequent reconstitution of NanoLuc, resulting in the emission of blue light upon conversion of the substrate (furimazine; $\lambda_{max}$ ~ 460 nm). Ratiometric detection is achieved by the addition of a calibrator luciferase, consisting of a tight fusion of NanoLuc to fluorescent acceptor mNeonGreen, which produces a green signal ($\lambda_{max}$ ~ 520 nm) upon substrate conversion through bioluminescent resonance energy transfer (BRET). The homogeneous solution-based assay is conducted by mixing the sample and sensor components with the NanoLuc substrate and recording the emission spectrum. In the absence of antigen (left) no split complementation occurs resulting in predominantly green emission from the calibrator luciferase (low blue/green ratio), while the presence of antigen (right) results in NanoLuc reconstitution and an increase in blue-light emission (high blue/green ratio). **b** Schematic overview of the antibody conjugation method. Fusion between the adapter protein G carrying the photoreactive non-natural amino acid p-benzoylphenylalanine and either large BiT or small BiT (Gx-LB and Gx-SB, respectively) enables covalent coupling to the constant Fc-binding domain of most IgG-type antibodies, including all human IgGs, by exposure to long-wavelength ultraviolet (UV) light ($\lambda = 365$ nm). Non-reducing SDS-PAGE analysis of the photoconjugation reaction (Rxn.) illustrates the successful coupling of primary antibody (Ab, 10 μM) to either one (indicated with 1×) or two (2×) protein G adapters (10 μM). Reactions were performed in PBS (pH 7.4) for 1 h on ice. Subsequent small-scale Ni$^{2+}$-affinity chromatography removed unconjugated antibody (FT, flow-through) resulting in pure antibody conjugates (Pur.). Gel image depicts a representative image from $n = 3$ independent experiments, with similar results. Source data is provided in Supplementary Fig. 9a.

and many other mammalian isotypes, this approach offers a general method for the rapid synthesis of antibody-luciferase conjugates from commercially available antibodies.

To establish proof of principle and characterize the performance of the RAPPID sensor platform, we first developed a sensor targeting cardiac troponin I (cTnI), an important protein biomarker for a myocardial injury that requires highly sensitive detection at picomolar concentrations (Fig. 2a)[30]. To this end, Gx-LB and Gx-SB were photoconjugated to a pair of commercially available anti-cTnI antibodies with distinct binding epitopes (19C7 (mIgG2b) and 4C2 (mIgG2a), respectively, Fig. 1b). Subsequently, increasing concentrations of cTnI were added to a reaction mixture consisting of 1 nM of both antibody-luciferase components and incubated for 30 min at room temperature followed by the addition of NanoLuc substrate. A maximal 281-fold increase in luminescence intensity was observed upon the addition of cTnI, with a detection regime spanning four orders of magnitude and a limit of detection of 4.2 pM (Fig. 2b and Supplementary Table 2). Control experiments show that the low background activity that is observed under these conditions in the absence of analyte (<0.5%) is primarily caused by the residual activity of uncomplemented LB and to a lesser extent by the intrinsic affinity between split fragments LB and SB (Supplementary Fig. 8a). These initial results demonstrate that target binding effectively stimulates NanoLuc complementation, resulting in a sensitive bioluminescent signal output with a high signal-to-background ratio.

The luminescent output follows a bell-shaped dose-response curve, as evidenced by a decrease in signal at cTnI concentrations

>10 nM (Fig. 2b). This "hook" effect is well-known and occurs because at high target concentrations the two sensor components no longer bind to the same target to form an enzymatically active ternary complex, but instead each bind to different target molecules[31,32]. To provide a means for understanding and tuning the response characteristics of the RAPPID sensor, we developed a thermodynamic model that analyzes the equilibria involved in the target-induced formation of the luminescent ternary complex (Supplementary Fig. 1 and Supplementary Fig. 2). We investigated critical parameters that can be used to modulate the sensor response and performed nonlinear fitting of the experimental data to the model (Fig. 2b). The model predicts that sensor sensitivity and response are strongly dependent on the antibody affinities and that the detection regime can be tuned by varying the concentrations of antibody-luciferase conjugates (Supplementary Fig. 3a, c). We confirmed the latter predictions by performing RAPPID assays at sensor component concentrations of 10, 1, and 0.1 nM respectively (Supplementary Fig. 8b, c). An additional parameter that affects sensor performance is the intrinsic affinity between split NanoLuc components LB and SB. As predicted by the model, attenuating the interaction between LB and SB from $K_d = 2.5$ μM to $K_d = 190$ μM does not allow effective luciferase complementation in the sandwich complex (Supplementary Fig. 3b). Experimentally, increasing the strength of the interaction ($K_d = 0.28$ μM) does not enhance luciferase complementation much further in the presence of the analyte, but does results in a higher background signal in the absence of analyte and therefore a smaller dynamic range (Supplementary Fig. 9b). These observations indicate that the thermodynamic

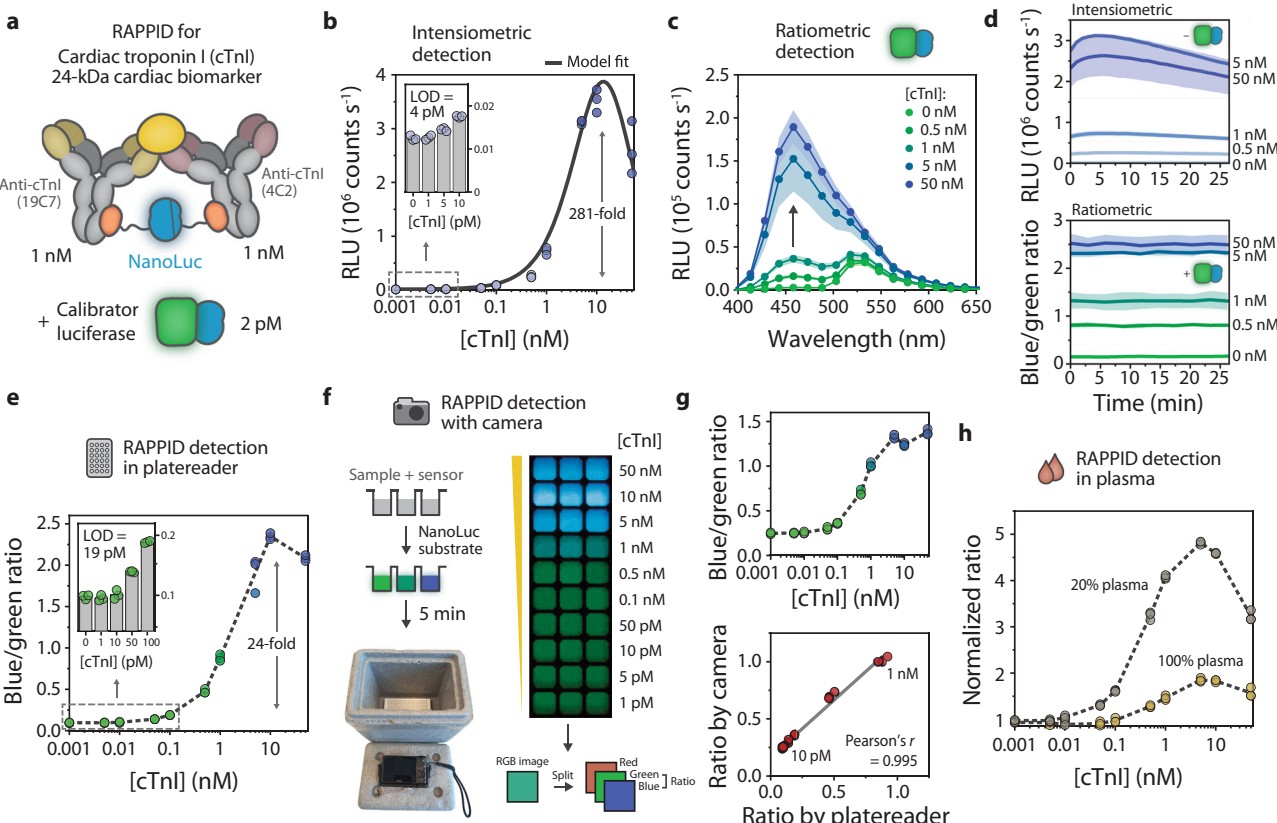

**Fig. 2 RAPPID enables robust ratiometric detection of cardiac troponin I (cTnI) in buffer and plasma and allows detection with a digital camera.**
**a** Schematic illustration of the sensor components used for the detection of cTnI. All experiments were performed in technical triplicates with independent preparations of the analyte, in buffer (PBS (pH 7.4), 0.1% (w/v) BSA), with 1 nM anti-cTnI (19C7) conjugated to Gx-LB, 1 nM anti-cTnI (4C2) conjugated to Gx-SB, and 2 pM calibrator luciferase, unless indicated otherwise. Reaction mixtures were incubated for 30 min at room temperature prior to the addition of NanoLuc substrate and recording of the emission spectra. The blue/green ratio for ratiometric detection was calculated by dividing bioluminescent emission at 458 nm by emission at 518 nm. **b** Intensiometric sensor response curve for cTnI. RLU, relative luminescence units. Inset shows sensor response at low target concentrations. Limit of detection (LOD) was determined at 4.2 pM with a maximal fold change of 281. An overview of all sensor characteristics including LOD confidence intervals and Z′-factor statistical coefficients can be found in Supplementary Table 2. Experimental data were fitted to a thermodynamic model (solid line, see Supplementary Fig. 4). **c** Bioluminescence spectra at various concentrations of cTnI in the presence of 2 pM calibrator luciferase. **d** Sensor output over time for the experiments described in (**b**) (top, intensiometric) and (**c**) (bottom, ratiometric). Data in (**c**) and (**d**) are represented as mean ± sd. of technical triplicates, with $n = 3$ independent preparations of the analyte. **e** Sensor response curve for increasing concentrations of cTnI. Blue/green ratios were calculated from the spectra in (**c**), and show a 24-fold maximal increase in signal. The inset displays sensor behavior in the low-concentration regime as indicated by the dashed rectangle, with a limit of detection (LOD) of 19 pM. **f** Bioluminescent emission does not require external excitation and therefore enables detection with a conventional digital camera (Sony DSC-RX100) inside a box (left bottom). Photograph of the sensor response for increasing cTnI concentration (right), taken 5 min after addition of the NanoLuc substrate. **g** Response curve calculated from the photograph in (**f**) (top) and the correlation between results from the plate reader and digital camera (bottom) for cTnI concentrations below 1 nM. A solid line indicates linear fit. Correlation analysis yielded a Pearson correlation coefficient ($r$) = 0.996. **h** Sensor response in 100% (yellow) and 20% (gray, diluted in buffer) human blood plasma, recorded in the plate reader. In (**b**), (**e**), (**g**), (**h**) individual data points are represented as circles, bars represent mean values, and dashed lines connect mean values. Data represent technical replicates, with $n = 3$ independent preparations of the analyte. Source data are provided as a Source Data file.

model can inform the behavior of RAPPID sensors and facilitate the rational design of their molecular characteristics.

An inherent property of intensiometric luciferase assays is depletion of the substrate over time and a concurrent decrease in absolute signal intensity, which hampers reproducibility and prohibits quantitative measurements. This is especially problematic for point-of-care applications, where it is often difficult to calibrate the assay and compensate for variations in substrate concentration, as well as environmental conditions such as pH, temperature, and ionic strength. We hypothesized that a calibrator luciferase based on NanoLuc, but with a shifted emission maximum, could be added to the sample together with the sensor components to provide a direct internal calibration by ratiometrically comparing two different emission wavelengths. To construct the calibrator luciferase, we fused NanoLuc to the green

fluorescent protein mNeonGreen (mNG-NL), resulting in green luminescent emission ($\lambda_{max} \sim 520$ nm) due to efficient BRET from NanoLuc to the fluorescent acceptor mNeonGreen[33]. The calibrator was added to the cTnI sensor mixture at a final concentration of 2 pM, after which luminescent spectra were recorded at various cTnI concentrations (Fig. 2c). In contrast to intensiometric detection, where the luminescent signal quickly reached a maximum and then decayed, the ratiometric signal was stable over a prolonged period of time, even at low substrate concentrations (Fig. 2d and Supplementary Fig. 10). A similar dose-response was observed, with a maximal signal at 10 nM cTnI and a limit of detection of 19 pM (compare Fig. 2b, e). Although the dynamic range is lower than in the intensiometric assay due to spectral overlap between blue and green emission, the >25-fold change in emission ratio is robust ($Z' = 0.95$, see Supplementary

Table 2) and greater than those typically seen in BRET-based sensors[10,15]. In addition, the calibrator concentration can be adjusted in order to obtain an optimal response at the desired detection regime of the analyte (Supplementary Fig. 11). In the experiments reported so far, antibody-luciferase sensor components were pre-incubated with the analyte to allow sufficient time for complex formation, followed by the addition of substrate and signal detection. The use of the calibrator luciferase additionally allows for a one-step assay in which sensor components and substrate are added simultaneously, and the formation of the sandwich complex can be monitored in real-time using ratiometric detection, even though the absolute intensity decreases because of substrate turnover. As expected, the kinetics of complex formation was found to depend on cTnI concentration, approaching equilibrium at ~60 min under these conditions (Supplementary Fig. 12).

We also applied the ratiometric nature of the RAPPID sensor to facilitate straightforward signal recording with a digital camera. Using identical reaction conditions as described for Fig. 2c, the cTnI concentration-dependent color change from blue to green was clearly observed in images recorded with a standard digital camera (Fig. 2f). The emission ratios were calculated from the blue- and green-color channels of the RGB image and showed a similar dose-response as in the plate reader assay, with a linear correlation for cTnI concentrations ≤1 nM (Fig. 2e, g). Additionally, we assessed sensor performance in human blood plasma and serum. The luminescent output signal was substantially suppressed in 100% plasma, which can be partially attributed to the absorption of blue light by hemoglobin and biliverdin (Fig. 2h and Supplementary Fig. 13)[34,35]. Nevertheless, the high intrinsic sensitivity of the RAPPID sensor still enabled ratiometric detection, with a reliable >fivefold change in emission ratio in both 20% blood plasma and serum (Fig. 2h and Supplementary Fig. 14). These results establish the RAPPID sensor platform as a robust ratiometric detection method with a large intrinsic dynamic range. As a result, in-solution measurements can be performed in clinically relevant media such as blood plasma, while facile signal recording with a digital camera should enable rapid diagnostic testing at the point of care.

**Target modularity and clinical validation**. To demonstrate that the RAPPID workflow can be implemented for alternative targets, we next developed a sensor for CRP, a pentameric protein with clinical relevance as an inflammation marker in cardiovascular disease[36], COPD[37], and infectious diseases (Fig. 3a). Two anti-CRP antibodies (C135 and C6) were photoconjugated to Gx-LB or Gx-SB, respectively, using similar conditions as described for cTnI, and SDS-PAGE was used to confirm efficient conjugation and purification (Supplementary Fig. 15). Using 1 nM of C6-LB, 10 nM C135-SB, and 2 pM calibrator luciferase and an incubation time of 30 min before addition of the substrate, an 18-fold increase in emission ratio was observed when varying the CRP concentration in the pM–nM regime, with an LOD of 2.9 pM (Fig. 3b). We confirmed complex formation within the incubation time by monitoring the sensor response over time, which demonstrated that a stable ratiometric signal was achieved rapidly and, for high analyte concentrations, within 15 min (Fig. 3c and Supplementary Fig. 16). We also used the CRP assay to assess the effect of temperature on assay performance, which is an important feature to consider when testing at the point of care. While the absolute intensity of both the sensor and the calibrator luciferase increased by 30–40% between 20 °C and 37 °C, the ratio of the two signals, and thus the RAPPID sensor output, remained stable in this temperature window (Supplementary Figs. 17 and 18).

Next, we benchmarked the analytical performance of RAPPID to a commercially available high-sensitivity ELISA, used in clinical care for cardiovascular risk assessment at low mg L$^{-1}$ levels in human blood plasma (~8–80 nM)[38,39]. A linear correlation (Pearson's $r = 0.996$) was observed between CRP concentrations as determined by ELISA and RAPPID (Fig. 3d and Supplementary Fig. 19), confirming the accuracy of our CRP sensor. Further validation was performed by comparing RAPPID to an automated immunoturbidometric assay used in the clinic for routine measurements of CRP as an inflammation marker in blood plasma at high mg L$^{-1}$ levels (~0.03–2.4 μM). Dilutions of individual, freshly collected 1-μL patient plasma samples (total $n = 40$) were measured using RAPPID and detected with a digital camera. In parallel, samples from the same patients were measured using the clinical assay (Fig. 3e and Supplementary Fig. 20). Our results showed excellent linear correlation and a small proportional difference with the clinical assay (Pearson's $r = 0.993$, slope $1.08 \pm 0.02$), and good reproducibility with an average coefficient of variation of $8 \pm 6\%$ derived from technical triplicates (Fig. 3f, g). Combined, these validation studies illustrate that RAPPID enables the accurate determination of biomarkers in a clinical setting. High assay sensitivity allows dilution of plasma samples, which further reduces potential matrix effects and allows tunable measurement of the analyte across a wide physiologically relevant concentration range. In addition, the "mix-and-measure" workflow substantially reduces total assay time and eliminates the need for multiple wash steps, opening up possibilities for point-of-care diagnosis.

**Antibody detection using RAPPID**. Our previously developed LUMABS BRET sensor platform enables sensitive and quantitative detection of specific antibodies but requires the genetic incorporation of well-defined, high-affinity epitopes or mimitopes, which are not always available[14]. The RAPPID platform provides assay formats for the detection of clinically relevant antibodies with complex, discontinuous epitopes and for the detection of heterogeneous antibody responses. We first developed sensors for ADAs, which are raised by the immune system against therapeutic antibodies upon repeated administration and are the main reason for rapid clearance[40]. We utilized our photoconjugation strategy to directly couple both Gx-LB and Gx-SB to three important clinical therapeutic antibodies: cetuximab which targets EGFR and is used for colorectal cancer treatment[41], and adalimumab and infliximab which target TNFα and are prescribed for a variety of autoimmune diseases[42] (Fig. 4a and Supplementary Fig. 21). Similar dose-response curves were obtained for all three ADAs (Fig. 4b). Kinetic analysis revealed that the addition of the calibrator luciferase is crucial for stable sensor output and that a constant ratiometric signal is achieved within 30 min after the addition of the substrate (Supplementary Fig. 22). Because both antibody-luciferase conjugates contain the same antibody that binds to the bivalent ADA target, a statistical mixture is expected to form that also includes a fraction of non-luminescent ternary complexes. Despite this, the RAPPID sensors exhibited a robust, five to sixfold change in emission and LODs between 59 and 81 pM (Supplementary Table 2).

Next, we applied RAPPID to detect antibodies that bind structurally complex, discontinuous epitopes, employing it to monitor therapeutic antibodies that target the TNFα homotrimer as proof of concept[42]. TNFα-binding therapeutic antibodies, such as adalimumab and infliximab, are widely administered anti-inflammatory drugs for the treatment of diseases like rheumatoid arthritis[43] and inflammatory bowel disease[44]. Therapeutic drug monitoring (TDM) of these antibodies is important in the clinic to achieve optimal treatment efficiencies and to reduce possible

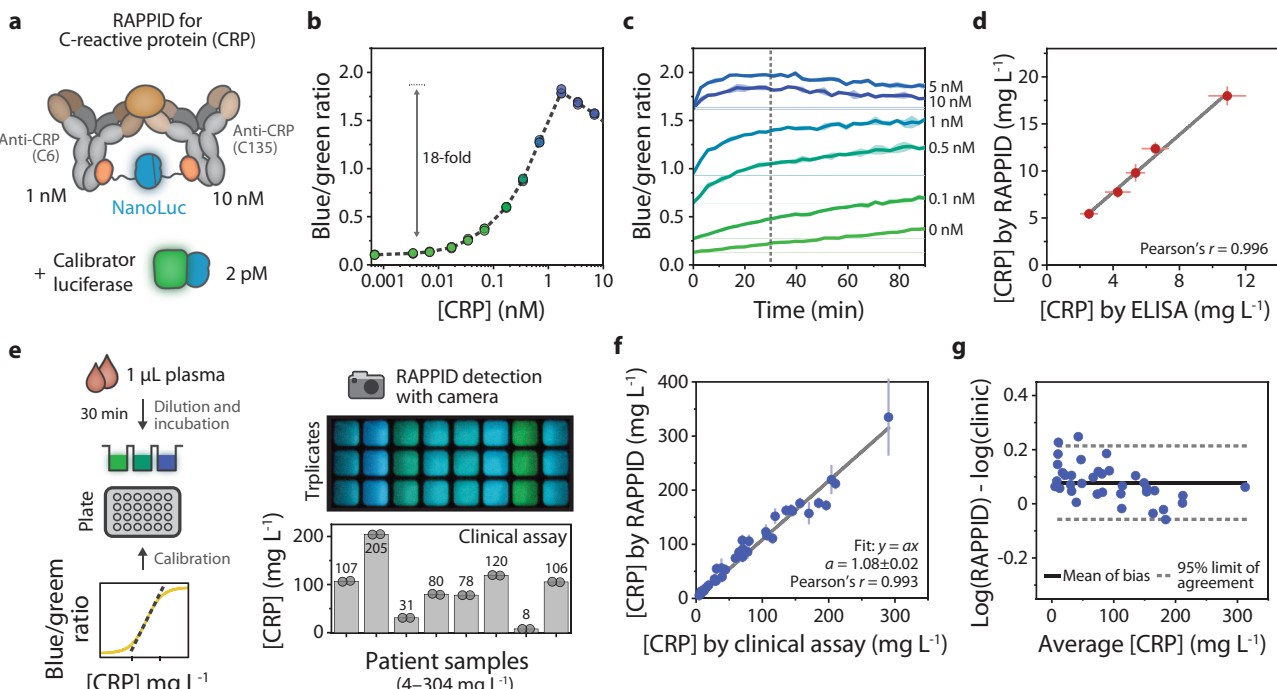

**Fig. 3 Development and clinical validation of a RAPPID sensor for the detection of C-reactive protein (CRP). a** Schematic illustration of the sensor components used for the detection of CRP. All RAPPID assays were performed in technical triplicate with independent preparations of the analyte, with 1 nM anti-CRP (C6) conjugated to Gx-LB, 10 nM anti-CRP (C135) conjugated to Gx-SB. Reaction mixtures were prepared in buffer (PBS (pH 7.4), 0.1% (w/v) BSA) and incubated for 30 min at room temperature prior to addition of NanoLuc substrate and recording of the emission spectra unless indicated otherwise. The blue/green ratio was calculated by dividing emission at 458 nm by emission at 518 nm. **b** Sensor response curve for increasing concentrations of CRP, displaying an 18-fold maximal increase in signal. Data points represent technical replicates, with $n = 3$ independent preparations of the analyte. **c** Sensor output over time at various concentrations of CRP, as indicated in the graph. All components (sensor, calibrator, analyte, substrate) were added simultaneously at $t = 0$. The dashed line indicates the 30-min. incubation time used in all other assays in this figure. Data are represented as mean ± sd of technical triplicates, with $n = 3$ independent preparations of the analyte. **d** Correlation of CRP concentrations measured by ELISA and RAPPID. Samples were prepared in diluted blood plasma and quantified using calibration curves obtained in buffer (see Supplementary Fig. 19). Error bars represent mean ± sd (technical replicates, with $n = 4$ independent preparations of the analyte). The solid line indicates linear fit. Regression analysis yielded a Pearson's correlation coefficient ($r$) = 0.996. **e** Clinical validation was performed against an agglutination-based clinical CRP assay with a detection range of 4–304 mg/L (0.03–2.4 μM). For RAPPID assays, samples were diluted 1000-fold or 4000-fold in a buffer prior to incubation, and detection was performed using a digital camera (for full method details, see Supplementary Fig. 20). The photograph shows the sensor output for eight patient plasma samples measured in triplicate, with corresponding CRP concentrations as determined by the clinical assay in the histogram below. Bars and numbers represent mean values. **f** Quantification of CRP levels in patient plasma, total $n = 40$. The graph shows CRP concentrations as determined by the RAPPID assay plotted against results from the clinical assay. Values for the clinical assay represent the average of $n = 2$ independent measurements. For RAPPID, values represent the mean ± sd of technical replicates from $n = 3$, wherein the patient samples were independently prepared. The solid line indicates a linear fit, with a slope of 1.08 ± 0.02 and Pearson's $r$ 0.993. Four samples yielded concentrations below the lower assay limit (<4 mg/L) in both the clinical assay and RAPPID assay, these were omitted in the statistical analysis. The underlying data are listed in Supplementary Table 3. **g** Bland–Altman analysis for the data described in (**f**). The solid line represents the mean of bias between RAPPID and the clinical assay, dashed lines indicate the 95% limit of agreement (mean ± 1.96 × sd). Source data are provided as a Source Data file.

side effects. We first applied the standard RAPPID format to detect infliximab using two anti-infliximab-luciferase conjugates, and a moderate three-fold change in emission was observed, possibly due to the formation of non-productive complexes as described above for the ADA sensors (Supplementary Fig. 23). To overcome this issue, we implemented an alternative RAPPID format by introducing a direct genetic fusion of SB to TNFα and combined it with a Gx-LB-photoconjugated type-2 anti-antibody, binding to the therapeutic antibody infliximab, or a type-3 anti-antibody, binding to the complex of the target therapeutic antibody adalimumab and TNFα (Fig. 4c and Supplementary Fig. 24). Because of the relatively high cut-off values and trough concentrations of adalimumab and infliximab[45–48], assays were performed using higher sensor concentrations (10 nM anti-antibody-LB and 100 nM TNFα-SB). As a result, we observed high substrate turnover and a quick decrease in luminescence intensity within a few minutes after adding the NanoLuc

substrate (Fig. 4d). The addition of the calibrator luciferase was therefore critical to allow accurate, time-independent antibody quantification, which resulted in robust changes in emission ratio in the clinically relevant therapeutic range, detected in diluted blood plasma using either a plate reader or a digital camera (Fig. 4e, Supplementary Figs. 25 and 26).

Kinetic experiments revealed that increasing the concentration of TNFα-SB markedly decreased the time to reach stable sensor output, from >30 min to less than 15 min for detecting 1 nM adalimumab or 1 nM infliximab (compare columns in Fig. 4f). At the high sensor concentrations used for the assays described in Fig. 4e (10 nM antibody-LB, 100 nM TNFα-SB), the formation of the sandwich complex is complete even within 5 min (Supplementary Figs. 25b and 26b). Collectively, these results illustrate that the RAPPID platform is highly versatile and that the standard assay format can be re-engineered to suit particular applications related to antibody detection, including fast, one-step

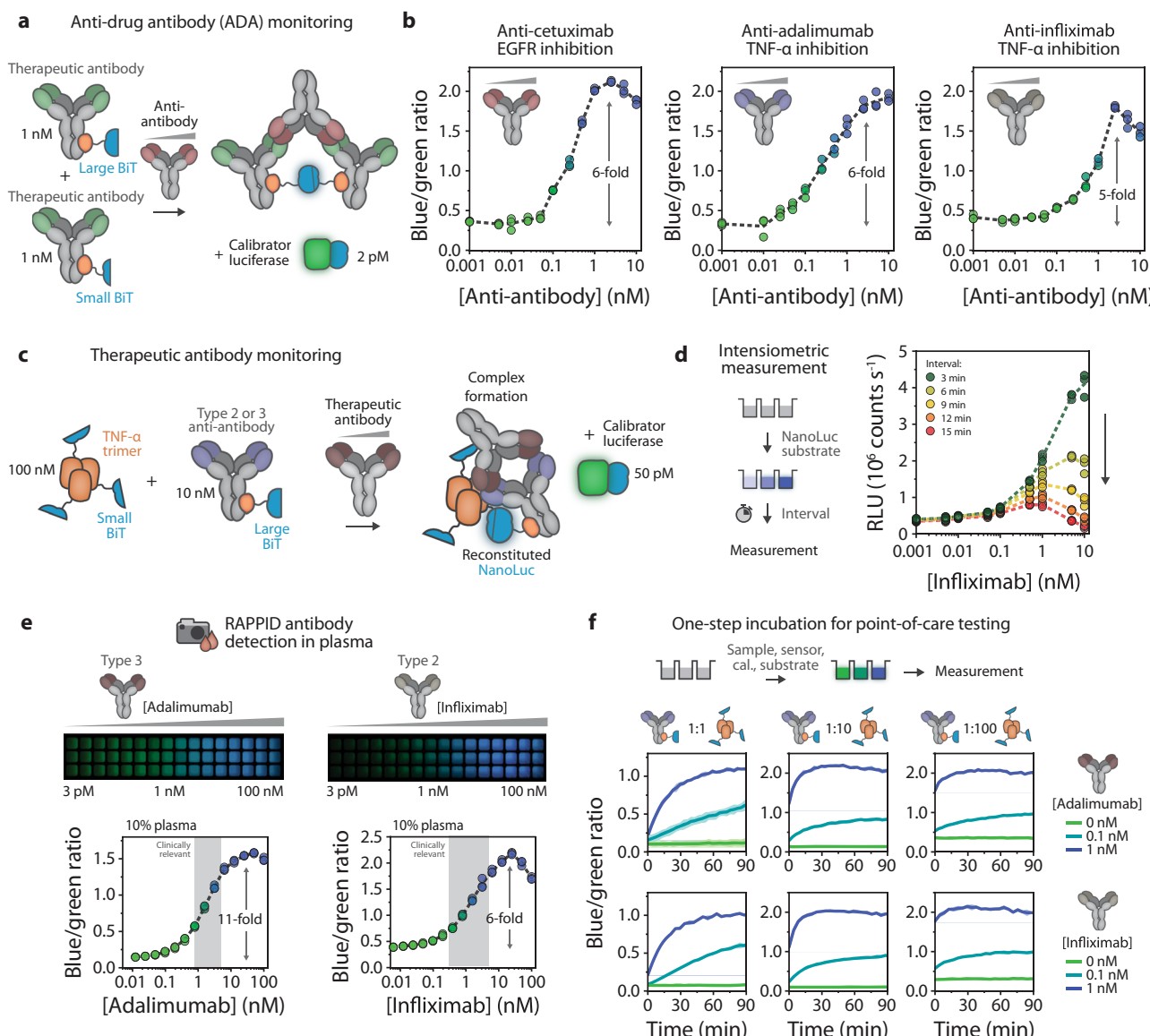

**Fig. 4 RAPPID enables monitoring of clinically relevant antibodies. a** Schematic overview of RAPPID sensors for the detection of anti-drug antibodies (ADAs). The use of a commercially available therapeutic antibody conjugated to both Gx-LB and Gx-SB enables binding of the ADA and the formation of a luminescent ternary complex. **b** Ratiometric sensor response curves in buffer (PBS (pH 7.4), 0.1% (w/v) BSA) of RAPPID sensors for anti-cetuximab, anti-adalimumab, and anti-infliximab, respectively. **c** Schematic overview of RAPPID sensors for detection of therapeutic antibodies adalimumab and infliximab using target antigen TNFα fused to SB, and the type 2 anti-infliximab and type 3 anti-adalimumab conjugated to Gx-LB. **d** Intensiometric detection of infliximab without calibrator luciferase. Luminescent measurements were performed at various intervals following the addition of the NanoLuc substrate. **e** Sensor response for adalimumab (left) and infliximab (right) detection in 10% plasma, diluted in the buffer. The same sample was used for detection with a digital camera (top) and the plate reader (bottom). Gray boxes represent the clinically relevant concentration range of each therapeutic antibody from cut-off value to trough concentration (8.5–53 nM and 3.3–47 nM for adalimumab and infliximab, respectively[45-48]), corrected for dilution. **f** Sensor response over time after one-step incubation of all assay components for both adalimumab (top row) and infliximab detection (bottom row), with varying ratios between sensor components antibody-LB and TNFα-SB, as indicated. Antibody-LB (1 nM) and TNFα-SB (1, 10, or 100 nM), therapeutic antibody (sample; 0, 0.1, or 1 nM), calibrator luciferase (cal.; 3–6 pM), and NanoLuc substrate (1000 × dilutions) were added simultaneously at $t = 0$. In (**b**), (**d**), (**e**) individual data points are represented as circles, and dashed lines connect mean values. In (**f**) data are represented as mean ± sd. All experiments were performed in technical triplicate, with $n = 3$ independent preparations of the analyte. Source data are provided as a Source Data file.

quantitative measurements of antibodies that bind discontinuous epitopes and therapeutic antibodies and their ADAs.

**RAPPID for SARS-CoV-2 antigen and antibody detection.** The modularity of the RAPPID platform is also attractive for the rapid development of sensors for newly emerging targets. The COVID-19 pandemic, caused by the SARS-CoV-2 virus, has highlighted the need for fast, low-cost, point-of-care viral detection to complement centralized PCR-based testing and vaccination[49–52]. Two recently developed monoclonal antibodies[53] that specifically target the immunogenic spike glycoprotein of the SARS-CoV-2 virus allowed us to develop RAPPID sensors for viral antigen detection. The antibodies (47D11 and 49F1) bind distinct epitopes on the spike protein which natively exists as a trimer. We constructed two sensor variants, using either both antibodies (heteropair, 47D11-LB with 49F1-SB) or only a single antibody (homopair, 47D11-LB with 47D11-SB) and directly utilized them

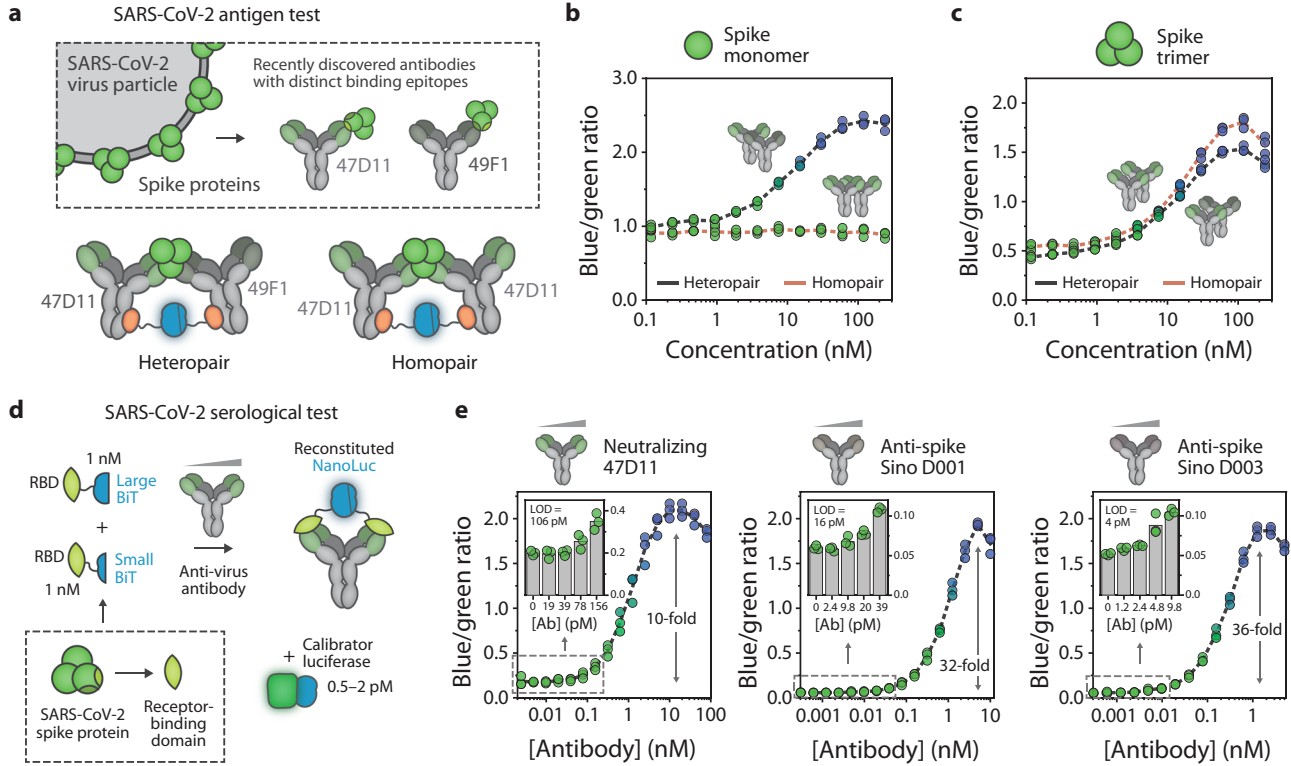

**Fig. 5 The RAPPID platform enables the detection of both the SARS-CoV-2 spike glycoprotein and anti-RBD antibodies. a** Schematic overview of RAPPID sensors for the detection of trimeric SARS-CoV-2 spike proteins. Recently discovered antibodies[53,64] targeting distinct binding epitopes of the immunogenic spike protein were used to generate sensors in either a heteropair or homopair setup. **b, c** RAPPID response curves in response to increasing concentrations of monomeric spike protein (**b**) and native trimeric spike protein (**c**) using both the heteropair (gray) and homopair (red) configuration. Photoconjugations were performed as described in Supplementary Fig. 27, and conjugation mixtures were used without further purification. A 200-fold excess of cetuximab was added to suppress background luminescence. Experiments were prepared with 1 nM anti-spike (47D11) conjugated to Gx-LB and 10 nM anti-spike (49F1 for heteropair, 47D11 for homopair) conjugated to Gx-SB. Subsequently, experiments were performed in buffer (PBS (pH 7.4), 0.1% (w/v) BSA) and incubated for 1 h at room temperature before addition of NanoLuc substrate and recording of the emission spectra. The blue/green ratio was calculated by dividing bioluminescent emission at 458 nm by emission at 518 nm. **d** Schematic overview of RAPPID sensors for direct monitoring of anti-virus antibodies against the receptor-binding domain (RBD) of SARS-CoV-2 spike protein. LB and SB were expressed as fusion proteins with the RBD, enabling detection of all RBD-targeting antibodies. **e** Ratiometric sensor response curves for neutralizing antibody 47D11 (left)[53], and commercially available anti-spike D001 (center) and D003 (right)[64]. Insets show sensor response at low target concentrations, which allowed calculation of the limit of detection, as indicated in the graphs. Experiments were performed with sensor and calibrator concentrations indicated in (**d**), prepared in buffer (PBS (pH 7.4), 0.1% (w/v) BSA), and incubated for 1 h at room temperature before addition of NanoLuc substrate. The fold increase is indicated in each graph. Individual data points are represented as circles, while dashed lines connect mean values. All experiments were performed in technical triplicate, with *n* = 3 independent preparations of the analyte. Bars in histograms represent mean values. Source data are provided as a Source Data file.

for sensing the spike protein without further purification (Fig. 5a and Supplementary Fig. 27). To scavenge non-conjugated Gx-LB and Gx-SB and consequently reduce the background signal in absence of the analyte, we added an excess of non-binding IgG to the assay solution. We hypothesized that we could differentiate between the monomeric and trimeric states of the spike protein, the latter being more likely to represent the active virus, by comparing the heteropair and homopair sensor response. Indeed, only the heteropair sensor variant, in which the antibody-luciferase conjugates are able to bind to distinct epitopes, exhibited a dose-response to monomeric spike protein, while no response was observed when using the homopair (Fig. 5b). Both sensor variants showed a concentration-dependent increase in emission ratio in response to the spike protein trimer, with a detection regime ranging between 1 nM and 100 nM (Fig. 5c). Model simulations suggest that the sensitivity could be further improved by using antibodies with higher affinities (Supplementary Fig. 3c). Indeed, when performing similar tests using commercially available anti-spike antibodies with higher affinities, the LOD could be decreased by almost an order of

magnitude, between 0.2 and 0.3 nM (Supplementary Fig. 28, 29 and Supplementary Table 2).

Containment of the ongoing pandemic and prevention of future outbreaks also critically depends on the development of serological assays as a tool to evaluate the extent of viral infection in the population and assess the efficacy of vaccine candidates. As such, we also developed a RAPPID variant for the detection of a wide panel of antibodies against the SARS-CoV-2 virus[54,55]. In a similar approach as described in the previous section, we adapted the standard RAPPID assay format by introducing the immunogenic receptor-binding domain (RBD) of the SARS-CoV-2 spike protein[56], directly fused to either SB or LB (Fig. 5d). The bivalent nature of antibodies promotes complementation of LB with SB, hence producing a bioluminescent signal upon RBD-LB and RBD-SB binding. After successful expression of RBD-LB and RBD-SB in HEK293T cells and subsequent purification (Supplementary Fig. 30), the sensor was assessed for its ability to detect three different anti-RBD antibodies: a recently discovered neutralizing antibody (47D11) and a commercially available ELISA pair (D001 and D003). Fig. 5e depicts the RAPPID

response curves of these antibodies, with LODs of 106 pM, 16 pM, and 4 pM for 47D11, D001, and D003, respectively, and a maximal change in emission ratio of 36-fold. As expected, the antibody (49F1) that binds the spike protein outside the RBD[53] did not induce a sensor response (Supplementary Fig. 31). The ratiometric detection enabled by the RAPPID assay is preferred over intensiometric detection of split NanoLuc complementation[57,58] as it allows quantitative measurements, which are important when studying the time-dependence of the antibody response and the correlation between antibody titers and immunological protection (for kinetic analysis, see Supplementary Fig. 32)[59,60]. Taken together, these results illustrate that the RAPPID platform allows fast development of quantitative assays for newly arising targets and that the modular design strategy enables multi-faceted viral detection of both SARS-CoV-2 antigen and antibodies.

Protein-based biosensors that generate a bioluminescent signal upon binding of a biomolecular target are attractive bioanalytical tools because they (1) enable highly sensitive detection based on enzymatic amplification, and (2) do not require external excitation, which renders them ideal for direct homogeneous detection in complex samples. In this study, we have established a comprehensive sensor platform based on antibody-guided split NanoLuc complementation combined with robust ratiometric output. The RAPPID platform is highly modular and broadly applicable as it allows the use of commercially available monoclonal antibodies as recognition elements. The required protein G-split NanoLuc fusion proteins are easily expressed in *E. coli* and obtained using standard affinity purification, while antibody conjugation is based on a one-hour illumination protocol that does not require expert laboratory equipment. Moreover, the use of a calibrator luciferase enables single-step, ratiometric detection of immunocomplex formation in real-time by monitoring the blue-to-green color change using a standard digital camera.

Combining these design aspects allowed the formulation of a straightforward workflow for developing in-solution immunoassays with low-pM sensitivity, high signal to noise, a large dynamic range, and excellent reproducibility for any relevant target, depending only on the availability of a suitable pair of antibodies. Sensors for traditional biomarkers such as cardiac troponin I and CPR, as well as newly emerging targets such as the SARS-CoV-2 virus, could be developed within days after antibody arrival. Depending on the affinity of the antibodies and the target architecture, robust changes in emission ratio ranging from 5 (for ADAs) to 25 were routinely obtained. The semi-flexible linkers that were employed are based on previous luminescent sensors and are known to effectively bridge the 10–15-nm distances encountered in sandwich immunoassays, and RAPPID assays were successfully developed for all antigens tested so far, representing a wide variety of architectures. Nonetheless, depending on the architecture of the sandwich complex, optimization of linker length and stiffness could be employed to further improve complementation efficiency for specific targets. Real-time monitoring enabled by the calibrator luciferase revealed that, as expected for a homogeneous assay, the kinetics of complex formation is affected by the concentrations of both the target and the antibody conjugates. However, in most cases, assay times of 15–30 min were sufficient to allow reliable quantification. Finally, the sensor's responsive range can be rationally adjusted to a specific application by varying the concentration of the antibody conjugates and the concentration of the calibrator luciferase.

The platform is versatile by design, which allowed us to create re-engineered RAPPID variants and multi-faceted detection assays, for example for measuring therapeutic antibodies and their anti-drug antibodies, as well as both SARS-CoV-2 antigens and antibodies. Additionally, the antibody-based sensor

components allow detection of targets with complex or discontinuous epitopes, which represents an important advantage compared to other bioluminescent sensors such as the LUMABS platform. A key aspect of the RAPPID platform is direct in-sample calibration by the addition of a calibrator luciferase that contains an identical luciferase domain but with a spectrally distinct emission profile. We took advantage of these features by developing assay variants for the detection of anti-TNFα and anti-RBD antibodies, in which either one or both parts of the split luciferases were genetically fused to TNFα and RBD, respectively. Similar approaches for the detection of anti-RBD antibodies were recently reported[57,58]. Although the ratiometric response is smaller than the reported intensiometric response, the use of a calibrator luciferase ensures that the ratiometric output signal is stable over time and less susceptible to experimental variation, rendering our assay more suitable for antibody quantification. We envision that the use of a calibrator luciferase can be applied as a general strategy to translate intensiometric luminescent assays into robust ratiometric assays with a time-independent and quantitative assay output. As such, other recently developed intensiometric sensors based on split luciferase complementation could benefit from a calibrator luciferase to further enhance their potential in a clinically relevant environment[21,58,61]. Furthermore, the concept can be extended by using different acceptor domains, such as quantum dots, organic fluorophores, or fluorescent protein domains, with a larger spectral separation from the blue light-emitting NanoLuc.

Ratiometric detection is also an important prerequisite for the successful application of bioluminescent assays in point-of-care diagnostics. Direct comparison of RAPPID with a commercially available ELISA kit and an automated clinical assay demonstrated that the blue-to-green ratiometric sensor signal can be detected fast and robustly with minimal equipment and low sample volumes while mirroring the clinically approved assays in terms of accuracy and sensitivity. In addition, implementation at the point of care can be envisioned by the integration of the bioluminescent sensor proteins into paper- or thread-based analytical devices[11,19]. To increase the sensitivity of such devices, RAPPID could also be combined with dedicated microfluidic devices, while dedicated readers could be developed that capture the bioluminescent light more efficiently than commercial digital or smartphone cameras[62]. A possible limitation of the current RAPPID assays for point-of-care applications is the limited long-term stability of sensor components at room temperature, in particular for the NanoLuc substrate. However, this could be addressed by packaging devices under low-oxygen conditions, or by applying a recently reported formulation of an all-in-one reagent that has a long shelf life at ambient temperatures[25].

In conclusion, RAPPID represents an attractive immunoassay format for point-of-care diagnostics, that combines the analytical performance of a laboratory-based immunoassay such as ELISA with the speed and ease of use of LFIAs. The possibility to integrate ratiometric bioluminescent assays such as RAPPID with (paper-based) microfluidic devices and the ability to develop a RAPPID assay for a new target analyte only within days, makes RAPPID an attractive immunoassay format for a broad spectrum of applications, including rapid screening methods, therapeutic antibody-drug monitoring, and the rapid detection of infectious diseases to be better prepared for future pandemics.

## Methods

**Cloning.** The pET28a(+) vectors containing DNA encoding for Gx-SB, Gx-LB, calibrator luciferase (mNG-NL), and SUMO-TNFα-SB were ordered from Gen-Script. The sequence of calibrator luciferase was reported by Suzuki et al.[33], and a His-tag at the N-terminus and a Strep-tag at the C-terminus were included to facilitate purification[27]. Site-directed mutagenesis to change SB sequences was

carried out using the QuikChange Lightning Site-Directed Mutagenesis kit (Agilent technologies) and specifically designed primers (Supplementary Table 1).

The RBD of the SARS-CoV-2 Spike glycoprotein (produced under HHSN272201400008C and obtained through BEI Resources, NIAID, NIH: Vector pCAGGS Containing the SARS-Related Coronavirus 2, Wuhan-Hu-1 Spike glycoprotein RBD, NR-52309) was genetically fused to either the LB or SB via a semi-flexible linker and subsequently cloned into the pHR-CMV-TetO2_3C-mVenus-Twin-Strep plasmid, a gift from A. Radu Aricescu (Addgene plasmid #113891)[63], by means of restriction-ligation approach. Snapgene 5.2.4 software was used for cloning design and sequence verification.

All cloning and mutagenesis results were confirmed by Sanger sequencing (BaseClear). The DNA and amino acid sequences of the fusion proteins are listed in the supplementary information (Supplementary Fig. 33–38).

**Protein expression**. The pET28a plasmid encoding for either Gx-LB or Gx-SB was co-transformed into *E. coli* BL21 (DE3) together with a pEVOL vector encoding a tRNA/tRNA synthetase pair in order to incorporate para-benzoyl-phenylalanine (pBPA)[28]. The pEVOL vector was a gift from Peter Schultz (Addgene plasmid # 31190). Cells were cultured in 2YT medium (16 g peptone, 5 g NaCl, 10 g yeast extract per liter) containing 30 µg/mL kanamycin and 25 µg/mL chloramphenicol. Protein expression was induced using 0.1 mM IPTG and 0.2% arabinose in the presence of 1 mM pBPA (Bachem, F-2800.0001). Following overnight expression at 20 °C, cells were harvested by centrifugation at 10,000 g for 10 min and lysed using the Bugbuster reagent (Novagen) and Benzonase (Novagen). Proteins were purified using Ni$^{2+}$ affinity chromatography followed by Strep-Tactin purification according to the manufacturer's instructions. Correct incorporation of pBPA was confirmed by Q-ToF LC-MS (WatersMassLynx v4.1), using MagTran v1.03 for MS deconvolution (Supplementary Fig. 7).

The pET28a plasmid encoding the calibrator luciferase fusion protein was transformed into *E. coli* BL21 (DE3). Cells were cultured in LB medium (10 g peptone, 10 g NaCL, 5 g yeast extract per liter) containing 30 µg/mL kanamycin. Protein expression was induced using 0.1 mM IPTG at 20 °C overnight. The harvested cells were lysed using the Bugbuster reagent and Benzonase. Proteins were purified using Ni$^{2+}$ affinity chromatography and Strep-Tactin purification.

The SARS-COV-2 spike trimer protein was expressed transiently in HEK-293T cells with a C-terminal trimerization motif and Strep-tag using the pCAGGS expression plasmid and purified from the culture supernatant by Strep-Tactin chromatography[53]. Similarly, the SARS-COV-2 spike monomer was expressed with a C-terminal Strep-tag and purified by Strep-Tactin chromatography[53].

The pET28a plasmid encoding SUMO-TNFα-SB fusion protein was transformed into *E. coli* BL21 (DE3). Cells were cultured as described above. The harvested cells were resuspended in Tris-HCl Buffer (40 mM, pH 8.0, containing 125 mM NaCl and 5 mM imidazole) and Benzonase. Cell disruption was then performed by ultrasonication on ice with 5 pulses of 20 s and an amplitude of 50%. After centrifugation, proteins were purified from the supernatant at 4 °C, using Ni$^{2+}$ affinity chromatography. The His-SUMO-tag was cleaved from TNFα-SB by adding 1:1000 (v/v) SUMO protease into the protein solution which was further dialyzed in Tris-HCl buffer (20 mM, pH 8.0, containing 150 mM NaCl and 1 mM DTT) overnight, at 4 °C. Cleaved TNFα-SB was obtained using Ni$^{2+}$ affinity chromatography to remove the cleaved His-SUMO-tag.

Lentiviral production of the two vectors, encoding either the RBD-LB or RBD-SB transgene, and subsequent protein expression was performed according to standard protocols as described in ref. [63]. Briefly, HEK293T cells (ATCC® CRL-3216™) were transiently co-transfected with the abovementioned transfer vectors, as well as the VSV-G envelope (pMD2.G) and packaging plasmids (pCMVR8.74). Plasmids pMD2.G and pCMVR8.74 were gifts from Didier Trono (Addgene plasmid #12259 and Addgene plasmid #22036, respectively). HEK293T cells were then infected in order to establish polyclonal stable cell lines via the integration of the lentiviral genetic elements into the cell genome. Cells were routinely cultured under standard culturing conditions (37 °C, 5% CO$_2$, 95% air, and 95% relative humidity) in Dulbecco's Modified Eagle Medium (DMEM 21885-025, ThermoFisher), supplemented with 10% Fetal Bovine Serum (FBS). For cell line maintenance, cells were split at a ratio of 1:5-1:10, upon reaching confluency. Protein expression was induced by the addition of 1 µg/mL doxycycline (D3447, Merck) in DMEM, supplemented with 2% FBS. Successful transduction of cells was monitored by fluorescence imaging of the reporter protein mVenus, ensuing induction. Protein-containing medium was harvested 4–5 days following induction and protein was purified by Strep-Tactin affinity chromatography. All the purified proteins were snap-frozen and stored at −80 °C until use.

**Photoconjugation**. Cardiac Troponin I antibodies 19C7 (4T21) and 4C2 (4T21), and CRP antibodies C6 (4C28cc) and C135 (4C28) were obtained from Hytest. Therapeutic antibodies cetuximab and infliximab were obtained via the Catherina hospital pharmacy in Eindhoven and Maxima Medisch Centrum pharmacy in Veldhoven, the Netherlands, respectively. Adalimumab (A1048-100) was obtained from Gentaur. Anti-adalimumab/TNFα monoclonal antibody (HCA207) and anti-infliximab (HCA213) were obtained from BioRad. The cDNA encoding the anti-SARS-COV-2 antibodies 47D11 and 49F1 were cloned into expression plasmids with human IgG1 heavy chain and Ig kappa light chain constant regions (Invivogen), containing the interleukin-2 signal sequence to facilitate secretion of the

antibodies. Subsequent transfection and expression were performed according to ref. [53] and protocols from InvivoGen in HEK-293T cells and purified using Protein-A affinity chromatography. Commercially available anti-SARS-COV-2 D001, D002, D003, and D004 were ordered from Sino Biological. Photoconjugation reactions were performed using a Promed UVL-30 UV light source (4 × 9 W). Samples containing antibody and Gx-LB or Gx-SB in PBS buffer (pH 7.4) were illuminated with 365 nm light for 30–180 min. The samples were kept on ice during photo-conjugation. If necessary, the photoconjugated products were further purified using Ni-NTA spin columns (ThermoFisher) followed by PD G10 desalting columns (GE Health) according to the manufacturer's instructions. Antibody-conjugates were stored at 4 °C or −80 °C until use. Uncropped and unprocessed SDS-PAGE gels can be found in the Supplementary Information or in the Source Data file.

**Luminescent assays**. Cardiac Troponin I (8T53) and CRP (8C72) were purchased from Hytest. Anti-cetuximab (HCA221), anti-infliximab (HCA213G) and anti-adalimumab (HCA205) were ordered from BioRad. Intensiometric assays were performed at sensor protein concentrations of 0.1–10 nM in a total volume of 20 µL PBS buffer (pH 7.4, 0.1% (w/v) BSA) in PerkinElmer flat white 384-well Optiplate. Measurements in diluted blood plasma and serum were performed by preparing the analytes in pooled human blood plasma (in ACD, DivBioScience) and human serum (male AB, Sigma Aldrich) and sensor proteins in PBS buffer (pH 7.4, 0.1% (w/v) BSA). After incubation of sensor proteins and analytes for 5–60 min, NanoGlo substrate (Promega, N1110) was added at a final dilution of 250–1000-fold. Luminescence intensity was recorded on a Tecan Spark 10 M plate reader with an integration time of 100 ms and data was collected using Tecan SparkControl v2.1. For ratiometric assays, an appropriate amount (0.5–50 pM) of calibrator luciferase was added into the samples and luminescence spectra were recorded between 398 nm and 653 nm with a step size of 15 nm, a bandwidth of 25 nm, and an integration time of 1000 ms. The blue/green ratio was calculated by dividing bioluminescent emission at 458 nm by emission at 518 nm. LOD and 95% confidence interval of the LOD were calculated in Microsoft Excel by linear regression of the response related to the analyte concentration for a limited range of concentrations (for an overview, see Supplementary Table 2).

The luminescence signal was also recorded by using a SONY DSC-RX100 digital camera. The plate was placed into a Styrofoam box to exclude the surrounding light. The pictures were taken through a hole in the box using the digital camera with exposure times of 10–30 s, F value of 1.8, and ISO value of 1600–6400. Method details of the clinical validation of the CRP-RAPPID assay can be found in Supplementary Fig. 20. CRP detection in the clinic was performed in plasma using a Atellica latex-enhanced immunoturbidimetric assay (random-access analyzer, Siemens Healthcare Diagnostics Inc. USA), which uses latex particles coated with anti-CRP antibodies that rapidly agglutinate in the presence of CRP, increasing the intensity of scattered light. The assays were performed according to routine protocols in the laboratory of clinical chemistry at the Rijnstate hospital in Arnhem, The Netherlands.

**Material availability**. Plasmids for recombinant expression of Gx-LB, Gx-SB, RBD-LB, RBD-SB, TNFα-SB, and calibrator luciferase mNG-NL will be made available through AddGene after publication. Protein and DNA sequences are available in the Supplementary Information. Antibodies 47D11 and 49F1 and the SARS-CoV-2 spike protein variants used as analytes are available upon request as specifically outlined in ref. [53].

**Ethics declaration**. All patient samples were obtained under protocols and guidelines approved by the local ethical committee of the Rijnstate Hospital (reference number: KCHL 2021-1790) and in accordance with the Declaration of Helsinki, and de-identified before use. All patients provided informed consent, and only samples were used in which CRP was ordered as part of standard clinical care and of patients who did not object against usage of their remnant blood for quality purposes.

**Reporting summary**. Further information on research design is available in the Nature Research Reporting Summary linked to this article.

## Data availability
The data generated to support the plots and findings within this study, including all raw luminescence data, are provided in the Source Data files with this paper. Source data are provided with this paper.

## Code availability
Custom-written code for the computer models and simulations that support the experimental findings in this study is available as Supplementary Software.

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

## Acknowledgements

We thank Dr. Leo van IJzendoorn and the members of the T.E.S.T. student team for help with the anti-adalumimab assay, Prof. Frank Grosveld (Erasmus MC, Rotterdam, The Netherlands) for providing access to the 47D11 and 49F1 antibodies, Dr. Maarten Broeren and Dr. Luc Derijks (Máxima Medical Center, Veldhoven, The Netherlands) for useful discussions regarding TNFα-binding therapeutic antibodies. We thank Nick Burlet for initial experiments on the CRP assay, Dr. Simone Wouters and Nick Burlet for help with the expression of the calibrator luciferase, and Pim de Vink for initial help with the thermodynamic model. This work was supported by the European Research Council, ERC starting grant (ERC-2011-StG 280255) and an ERC proof of concept grant (ERC-2016-PoC 755471), funding by the TU/e COVID19 University Fund, grants from the Netherlands Organization for Scientific Research, NWO-Take-off-1 grant (NWO, 17820), and RAAK.PRO Printing makes sense (RAAK.PRO02.066).

## Author contributions

Y.N., B.R., and E.v.A designed the study, performed experiments, developed the thermodynamic model, analyzed the data, and wrote the manuscript. E.H. developed and characterized RAPPID assays for TNFα antibodies and ADAs. L.B. contributed to the spike protein assays. A.M.P. developed the expression of the RBD-NL fusion proteins, B.T., C.V., and S.R. contributed to initial experiments for various RAPPID assays, R.A. supervised early experiments for this work that provided proof of principle, W.L. expressed and purified spike proteins and antibodies. B.B. and F.v.K. supervised the spike protein work and provided feedback on the manuscript. T.d.G. supervised the development of the thermodynamic model and provided critical feedback on the manuscript. M.v.B. supervised clinical validation work and analyzed the resulting data. M.M. conceived, designed, and supervised the study, analyzed the data, and wrote the manuscript. All authors discussed the results and commented on the manuscript.

## Competing interests

Y.N., Dr. Leo van IJzendoorn, and M.M. filed a patent application on 22 June 2020 on RAPPID and the ratiometric detection of luciferase assays using a calibrator luciferase (The Netherlands patent application PCT/NL2020/050406; patent applicant: Eindhoven University of Technology). The remaining authors declare no competing interests.
