## [Peer Review File · Nature Communications]

Reviewers' Comments:

Reviewer #1:

Remarks to the Author:

The authors present a new configuration of immunoassay using the split-luciferase NanoLuc. By constructing various formats, they demonstrate the broad utility of the approach to a range of analytes. The development of the modified Protein G-luciferase reagents is noteworthy, as these can be quickly adapted to many different applications. Demonstrating the value of using a calibrating luciferase is an enabling strategy; such an approach should be adopted by ALL assays using split (or whole) luciferases.

Concerns that should be addressed:

The development of the broadly-applicable Protein G-luciferase adaptors is impactful, as this eliminates the need for specialized reagents for many different analytes. However, it is not clear (from the manuscript) what the specific advantages are of using this conjugation of primary antibodies approach as opposed to secondary antibodies conjugated with luciferase fragments (Reference 24).

The authors seem to have over-simplified the ease of developing and implementing their RAPPID assay. The first step "selection of a pair of (commercial) antibodies" is only applicable to some of the assays presented in the manuscript. Specifically, anti-TNF therapeutic antibody monitoring (4C) and direct antibody (anti-cov-2) monitoring (4F) both required the genetic engineering of fusion proteins, neither of which are antibodies nor commercially available. The second step to crosslinking protein G-luciferase appears straightforward; however, this reagent is recombinantly expressed with a non-natural amino acid (and therefore arguably not 'straightforward'). The third step of "addition of both antibody-luciferase conjugates, the calibrator luciferase and the Nanoluc substrate to the sample, followed by detection" is somewhat misleading as all of the assays described have a 2-step incubation step (various times), followed by addition of substrate (various concentrations). The assay is justifiably much simpler to implement (once created) than many other platforms, but the over-simplification here is distracting.

The term 'therapeutic window' is generally understood to mean the range of doses/concentrations that are therapeutic, without causing toxicity. The lower limit, or trough concentrations, for TNF inhibitors varies depending on indication (Aliment Pharmacol Ther. 2020 Mar;51(6):612-628.). From reference 51 of the manuscript, the trough concentrations for infliximab are 3-7 µg/mL, which would be ~20-47 nM. Even these trough values are above the grey-boxed area labeled as "therapeutic window" in Figure 4E. As such, the assay appears to be more calibrated with sub-therapeutic concentrations than to clinically relevant levels.

The majority of assays have a downward trend in luminescence or ratio at higher analyte concentrations. For example, (Figure 4G, Supplementary Figure 8A) signals increase to a max of ~10 nM, but decrease above that. Is this a result of substrate depletion? What amount/concentration of NanoLuc substrate was used? Although not indicated specifically, results appear to be from an 'end-point' read of luminescence values, instead of a kinetic evaluation (eg initial velocity). Would using the latter allow for higher concentrations of analyte to be assessed?

The authors indicate that plasma concentration negatively impacts the luminescence (Supplementary Figure 11). Why was plasma used instead of serum? Does the fibrinogen present in plasma negatively affect the assay? In relation, the amount of plasma in each different assay is not clearly indicated. Was the dilution for each assay consistent? Including these parameters (substrate concentration/dilution, incubation time, run time, etc) would be a simple and valuable addition (perhaps added to the Supplementary Table).

In assays presented in Figure 4 C, authors state that "Type 2 or 3 anti-antibody" are used. Which type was used for each of the assays? How does the selection of these respective antibodies change the kinetics of the assay, and therefore affect the optimal read-time? Also, the intrinsic activity of Large BiT, as well as auto-complementation appear to be major limiting factors. Do the authors have insight into potential solutions, or is the background luminescence unavoidable?

When selecting a pair of antibodies that bind a single antigen, does the distance between respective binding epitopes of the antibodies make a difference in signal? Are there limitations to distance? For configurations that require a single antibody to simultaneously bind two different target antigens (direct antibody monitoring, Figure 4F), what are the limitations in size or orientation of the analyte?

Protein G conjugation of small BiT to antibodies does not seem to go to completion. Do the authors have any insight into why this is the case (Supplementary Figure 17,18) and how it could influence variability in the read-out of the assays?

It is surprising, and unacceptable, that there is no statistical analysis of any of the results. The only exception is Figure 3C, indicating Pearson's correlation.

Reviewer #2:

Remarks to the Author:

In the present manuscript, Ni et al described a novel immunoassay approach to detect a variety of analytes by coupling antibodies with a luciferase complementation method. This is a very interesting manuscript and well-presented manuscript and while this has similarities with recently described work see (Hwang et al, 2020) the innovation of this approach is in coupling with an internal luciferase 'calibrator' to improve the relative stability of the assay through ratiometric detection and as the authors demonstrate the relative adaptability of this approach to a number analytes in multiple formats. I will be of interest to see which of the potential benefits and future uses and of these approaches discussed by the authors come to fruition.

I have a number of additional comments for the authors to consider.

Comments:

While noted in the results (In 109) it would be useful for the authors to comment of the potential effect of bi-conjugation of the NanoBiT fragments to the antibodies and if this would affect the results. Particularly which respect to the LgBiT fragment which can have detectable luminescence without complementation.

The authors state that that the data represent triplicate experiments. It would be useful to know if this was three independent experiments or the one performed in triplicate. Furthermore, there appears to be significant variation in some assays (see Figure 2 C/D), which appears random between doses. Indeed, a key conclusion by the authors is that the "ratiometric signal is stable over time and less susceptible to experimental variation" and that the future use of these assays as point-of-care diagnostics that would need to be highly reproducible between samples. An analysis of the reproducibility eg (Z' factor or similar) of one the assay formats would support such claims of utility.

On page 6 the authors benchmark the RAPPID assays with current ELISA approaches and report similar/more sensitive detection with the RAPPID approach (pM-nM) v (8-80nM) however the data presented in supplementary Figure 13 appear to show that that RAPPID assays is 1000-fold less sensitive than ELISA (mg/L v ug/L). Is there are technical reason for this? In addition, it is not clear why the RAPPID approach is appears to be detecting almost twice as much CRP compared to ELISA in Figure 3C presumably from the same starting concentrations? Clarification of this would be useful.

Throughout the manuscript the Ni et al use SB/SmBiT to indicate the small fragment of Nluc (VTGYRLFEEKES) with an affinity of Kd 2.5 uM (peptide 101, Dixon et al 2016). Current nomenclature from the manufacture and that which prevails in the literature is that peptide 114 (VTGYRLFEEIL, Kd 190 uM, Dixon et al 2016) is designated SmBiT. Standardisation of this in the manuscript would assist the reader. In addition, for clarification in SuppFig22 sequences should be labelled SB2 and SB3 as per SuppFig9

Minor comments:

In Figure 2b is insert up to 100pM or 10pM?

There is no mention of Supplementary Figure 9 in the manuscript and would be useful to have inserted on page 4 lines 165-166 to support the statement.

Can more detail as to how the LOD was calculated be provided in the methods, and as above whether this was calculated from a single experiment performed in triplicate or from three separate experiments. Notably, the LOD seems to be variable between the cTnI assays Fig2b (LOD

4pM) and SuppFig8a where there does not appear to be a detectable signal at 5pM?

Reviewer #3:

Remarks to the Author:

In the presented manuscript Merkx and colleagues continue their long-term search for technologies that would allow rapid development of protein biomarker quantification assays. In this study they depart from their earlier efforts to build fully integrated bespoke biosensors systems and, instead, develop a molecular tool kit and its application protocols that allows rapid immune assay development for a broad range of protein analytes.

A person familiar with the earlier research outputs of the group can see the evolution of ideas that led to the current development. In the present study the authors apply the idea of ratiometric bioluminescence sensing, that was originally developed for BRET sensors, to split enzyme systems thereby getting around the problem of time resolved changes in emission that complicates measurements using Luciferase -based systems. They also borrow from the field of chemical biology in order to rapidly produce covalently linked antibody- reporter conjugates through crosslinking of genetically incorporated photoactivatable amino acid. They analyse in detail the optimal ways to minimise the "hook effect" – an important issue that arises from the use of non-cooperative target binders. They test the developed assay on a range of targets and provide data on its performance when monitoring the assay using simple consumer camera.

From the technical perspective the paper combines, improves, and optimises a range of pre-existing technologies to develop a new assay format. While these are interesting and well executed the novelty is not very high and this shifts the attention to the potential practical utility and impact of the developed platform.

The authors state that the platform "provides an attractive, fast, and low-cost alternative to traditional immunoassays, both in an academic setting, in clinical laboratories and for point-of-care applications."

In my opinion an improvement to the cost structure for the assays performed in academic laboratories, constitutes a modest advancement and does not need to be evaluated in detail. I do not think the authors made a convincing case that the presented technology has the potential to either replace or augment existing commercial solutions used by diagnostic laboratories. While the contemporary chemical pathology is using a lot of legacy technologies, the industry had decades and billions of dollars to optimise their use and to achieve very impressive accuracy and efficiency. For example a good comparison case is the CEDIA system (Thermo-Fisher) that is based on split beta-galactosidase. Given that the technology stems from 1980th it is using less elegant solutions than that in the present study but the fact that it is a clinically used assay platform suggests that the robust performance has been achieved.

Potentially, one can argue that the developed technology allows hardware downgrade by replacing complex immunochemistry stations with less sophisticated devices. However, unlike absorbance-based chemistry analysers, homogeneous luminescence assays are lot less frequently used in diagnostic practice and hence the existing hardware platforms can not be used. While, based on the evidence provided by the authors, their replacement may be technically and commercially possible it would require regulatory approval of a new class of medical devices which is cumbersome and is likely to have limited appeal to their users.

I think a well-argued case needs to be made to support the utility of the developed technology for the laboratory testing (see below) and at least the back of the envelope calculations would be needed to support authors assertion.

Point-of-Care. As COVID-19 pandemic convincingly showed development of rapid and cheap distributed biochemical diagnostics is yet an unsolved problem. At present there is no clear technological leader there as all, so far, reported approaches have their limitations. The authors are the leaders in this field and therefore should be able to comprehensively and critically assess the potential of the developed technology for PoC applications.

In the view of the above here are the issues that the authors should try to address in the revised version of the manuscript:

1) Assay format. At present the assay is composed of a range of reagents that also includes NanoGlo (furimazine) substrate from Promega of an unknown concentration. How stable is this substrate and in what format would it be utilised in the PoC assays?

2) Pre-incubation times and the total duration of the assay. These are very important parameters that define technical implementation of the test and its clinical utility as, for an example, in many countries doctor visits are short (10-15 min) and the data obtained outside of this time window is less competitive with the established diagnostic methods. Further, longer measurement times indirectly influence the accuracy of the test as at prolonged incubation leads to variability due to sample drying, temperature fluctuations, user-inflicted environmental changes etc. Furthermore, the requirement for two step assay (Step1: sample+ antibody conjugates, Step 2: NanoLuc substrate addition and emission measurement) also significantly complicates design of the test's hardware.

The Results and M&M sections refer to pre-incubation times of sample with the protein components of 5-60 minutes. The authors need to provide a time dependent activation data for at least couple of assays for unrelated analytes where they mix the sample and then withdraw aliquots at different incubation times followed by addition of the NanoGlo and data collection. The analysis of the dynamic ranges and CV at different incubation times should be instructive in estimating of the minimal time this assay can be performed at.

Finally, one would expect that antibodies would bind to their targets in seconds if not faster. If extended incubations are required than what step of the process do they support and what can be done to shorten this time.

3) Assay temperature sensitivity. How sensitive the assay to the temperature fluctuations? This is a critical parameter in PoC testing and while it can be controlled it comes at increasing cost and complicates technical implementation. This may be related to the point 2.

4) Assay stability and comparative performance. The authors argue that the assay can be used to measure clinical biomarkers in human samples albeit with significant sample dilution. However, they do not provide any data on performance of the assays in the actual clinical samples and their benchmarking against currently used diagnostic methods. For instance, the data they provide on cardiac troponin is very interesting as reliable detection of cardiac troponin at lower concentrations is an unsolved problem in diagnostic industry (For instance : Outliers as a cause of false cardiac troponin results: investigating the robustness of 4 contemporary assays. Clin Chem . 2011 May;57(5):710-8. doi: 10.1373/clinchem.2010.159830 and Improved Diagnostic Performance of High-Sensitivity Cardiac Troponin Assays Is an Artifact of Censored Data. Clin Chem, 2016, 62(12):1654-1657. doi: 10.1373/clinchem.2016.262634)

Benchmarking this assay against currently used diagnostic methods using real clinical samples even in solution format would significantly strengthen author's case and also test the assay's robustness against patient-originated experimental noise.

5) Can the assay be dried down and combined with blood filtering solutions that the authors used in their 2018 Angewandte Chemie paper? This would bring the system lot closer to a PoC format. The discussion of these aspects would be important.

6) Assay inhibition by serum. Authors show that plasma has inhibitory effect on the assay and make suggestions of the causes. This is a very important aspect as dilution both complicates the assay format particular for PoC applications and also reduces the actual assay's sensitivity. Understanding the origin of the inhibition would be important for assay optimisation. The authors conjecture that absorption of the blue emission may be one of the reasons for inhibition. Is this the primary reason? If so the reported variants of NanoLuc with long wavelength emission would be able to solve the problem.

Response to the reviewers

We were pleased that all reviewers appreciated our work and would like to thank them for the time and effort in assessing the manuscript and their constructive feedback. As requested by the reviewers we have performed extensive additional work including additional kinetics experiments and experiments that further elaborate on the advantages of using ratiometric detection to mitigate matrix, temperature and sample-to-sample variation. Most importantly, in collaboration with the clinical chemistry laboratory of the Rijnstate hospital, we have clinically validated our RAPPID assay for CRP detection on 40 individual patient samples, showing excellent correlation between the RAPPID assay (performed with a digital camera) and the automated, agglutination-based CRP assay used in the hospital. Please find a point-by-point response below to all the questions and comments of the 3 reviewers.

Reviewer 1

The authors present a new configuration of immunoassay using the split-luciferase NanoLuc. By constructing various formats, they demonstrate the broad utility of the approach to a range of analytes. The development of the modified Protein G-luciferase reagents is noteworthy, as these can be quickly adapted to many different applications. Demonstrating the value of using a calibrating luciferase is an enabling strategy; such an approach should be adopted by ALL assays using split (or whole) luciferases.

We fully agree, also regarding the value of using a calibrator luciferase as a generic, enabling strategy to make intensimetric luciferase-based assays much more robust and quantitative. Recently, several high-impact publications appeared using split-luciferase complementation, which all stand to benefit from using a calibrator luciferase: Elledge et al. (2021) Engineering luminescent biosensors for point-of-care SARS-CoV-2 antibody detection *Nature Biotechnology* 3688; Hall et al. (2021) Toward a Point-of-Need Bioluminescence-Based Immunoassay Utilizing a Complete Shelf-Stable Reagent *Anal. Chem.* 93, 5177-5184; Quiajano-Rubio et al. (2021) De novo design of modular and tunable protein biosensors *Nature* 591, 482-48. The latter already refers to our work and we refer to all these studies in the conclusion and discussion. (“Similar approaches for the detection of anti-RBD antibodies were recently reported^{58,59}. Although the ratiometric response is smaller than the reported intensimetric response, the use of a calibrator luciferase ensures that the ratiometric output signal is stable over time and less susceptible to experimental variation, rendering our assay more suitable for antibody quantification. We envision that the use of a calibrator luciferase can thus be applied as a general strategy to translate intensimetric luminescent assays into robust ratiometric assays with a time-independent and quantitative assay output. As such, other recently developed intensimetric sensors based on split luciferase complementation could benefit from a calibrator luciferase to further enhance their potential in a clinically relevant environment^{21,59,62}.”)

Concerns that should be addressed:

The development of the broadly-applicable Protein G-luciferase adaptors is impactful, as this eliminates the need for specialized reagents for many different analytes. However, it is not clear (from the manuscript) what the specific advantages are of using this conjugation of primary antibodies approach as opposed to secondary antibodies conjugated with luciferase fragments (Reference 24).

The main benefit of direct conjugation of the split luciferases to the primary antibodies is that split luciferase complementation is generally more efficient, because the sandwich complex that is formed is less complex and the average distance between the luciferase parts in this sandwich complex is smaller. Consequently, this results in a higher effective concentration and more efficient complex formation. It is difficult to make this quantitative, but with the RAPPID system we routinely observe a 20-200-fold increase in luminescence intensity, whereas the dynamic range of the systems reported in ref. 24 is typically 5-10 fold. With a more limited number of assay components, formation of the sandwich immunoassay is also likely to be faster and cheaper. We added the following sentence in the first paragraph of the results to clarify this further. (“To establish a universally applicable sensing platform, we employed protein G-based photoconjugation for the synthesis of sensor components. This enables direct conjugation to most commercially available primary antibodies, obviating the use of labeled secondary antibodies which increases complexity and can decrease the efficiency of sandwich complex formation²⁴.”)

The authors seem to have over-simplified the ease of developing and implementing their RAPPID assay. The first step “selection of a pair of (commercial) antibodies” is only applicable some of the assays presented in the manuscript. Specifically, anti-TNF therapeutic antibody monitoring (4C) and direct antibody (anti-cov-2) monitoring (4F) both required the genetic engineering of fusion proteins, neither of which are antibodies nor commercially available.

We agree that the statement regarding the ease of developing the RAPPID assay applies mostly to the format in which an orthogonal pair of (commercially available) antibodies can be used. Please note that also for antibody detection classical RAPPID assays can be developed (as demonstrated here in Figure 4B for anti-Cetuximab, anti- Adalimumab and anti-Infliximab), but the fold increase in luminescent emission is somewhat lower compared to the assays containing two orthogonal antibodies. The assays reported in Figure 4C (anti-TNF α therapeutic antibodies) and 5D (anti-SARS-Cov2 antibodies) represent variants of the original RAPPID assay that indeed involves fusion of one or both of the split luciferases to a target antigen, which requires some additional engineering of fusion proteins. These variations on the RAPPID format provide for a more robust signal increase and further expand the scope and thereby illustrate the added value of ratiometric bioluminescent detection. We have adjusted the text of the manuscript to more clearly distinguish between the different assay formats and provide a balanced discussion of the ease-of-use. (“In case of antibody detection, the dynamic range of the assay and its scope can be further increased with an evolved set of RAPPID variants that use genetic fusions of the split luciferase components to the antibody’s antigen.” and “We first applied the standard RAPPID format to detect infliximab using two anti-infliximab-luciferase conjugates, and a moderate three-fold change in emission was observed, possibly due to formation of non-productive complexes as described above for the ADA sensors (Supplementary Fig. 23). To overcome this issue, we implemented an alternative RAPPID format by introducing a direct genetic fusion of SB to TNF α , and combined it with a Gx-LB-photoconjugated type-2 anti-antibody, binding to the therapeutic antibody infliximab, or a type-3 anti-antibody, binding to the complex of the target therapeutic antibody adalimumab and TNF α (Fig. 4c and Supplementary Fig. 24).” and “We took advantage of these features by developing assay variants for the detection of anti-TNF α and anti-RBD antibodies, in which either one or both parts of the split luciferases were genetically fused to TNF α and RBD, respectively.”)

The second step to crosslinking protein G-luciferase appears straightforward; however, this reagent is recombinantly expressed with a non-natural amino acid (and therefore arguably not ‘straightforward’).

While the incorporation of a non-natural amino acid may not seem straightforward, in this case the expression and purification of the recombinant photo-crosslinkable protein G-luciferases is similar to normal recombinant protein expression in *E. coli*. The only difference is that in addition to the expression plasmid for the fusion protein, a second plasmid encoding for the tRNA and tRNA-synthase needs to be co-transformed. Bacteria containing both plasmids can be selected by using two antibiotic selection markers instead of one. The plasmid that encodes for the tRNA synthetase/tRNA system was obtained via AddGene, and the expression plasmids for the Gx-LB and Gx-SB fusion proteins will also be made available via AddGene. Recombinant expression of the protein with the non-natural amino acid is the same as normal recombinant expression in *E.coli*, except that the non-natural amino acid pBPA (which is commercially available) is added to the culture medium. The presence of His- and Strep-tags at the N- and C-termini of the protein allows purification of the full-size protein using two affinity purifications steps, that do not require sophisticated HPLC or FLPC systems but are routinely performed via simple gravity flow columns. The expression yields of both proteins are also good and similar to standard protein expression (~18 mg/L for Gx-LB and ~5 mg/L for Gx-SB). Any laboratory with basic infrastructure for recombinant protein expression in *E. coli* can make sufficient proteins in only a few days to do thousands of assays. To ensure the wide dissemination and adaptation of the RAPPID system, we have added a dedicated step-by-step protocol to the Supplementary information that not only describes the production of the fusion proteins, but also provides a detailed instruction for the photo-crosslinking reactions.

The third step of “addition of both antibody-luciferase conjugates, the calibrator luciferase and the Nanoluc substrate to the sample, followed by detection” is somewhat misleading as all of the assays described have a 2-step incubation step (various times), followed by addition of substrate (various concentrations). The assay is justifiably much simpler to implement (once created) than many other platforms, but the over-simplification here is distracting.

In most of the experiments we indeed first pre-incubated the sample with the antibody-luciferase conjugates to allow complete complex formation, followed by addition of substrate and signal detection. However, one-step detection is also possible, in particular because the presence of the calibrator luciferase allows reliable quantification of the sandwich complex over a prolonged time during which substrate concentration (and thus absolute signal intensity) decreases. We have performed several additional experiments in which we added all components simultaneously, which allowed us to study the kinetics of sandwich complex formation (Supplementary Fig. 10, 12, 16, 22, 25, 26 and 32). These results confirmed that ratiometric detection indeed allows reliable monitoring of complex formation, whereas the time dependence of the bioluminescent intensity is dominated by the decrease in substrate concentrations (compare e.g. Figure 3C and Supplementary Fig. 16, and see Supplementary Fig. 10, 12, 22, 25, 26 and 32). We saw that even when the absolute intensity has decreased to <5%, the ratio between green and blue luminescence remains stable and is thus a reliable measure of complex formation. As expected, the kinetics of complex formation depend on the specific assay conditions, such as the kinetics of the antibody-antigen interaction and the analyte concentration. For some RAPID assays, complex formation is essentially complete in 5-10 min, such as for the adalimumab and infliximab assays that use relatively high concentrations of the sensor components (Supplementary Fig. 25 and 26). We have added some of these new data to figures in the main text (i.e. Figure 3C and 4F), while others are shown in the Supplementary information. The kinetics of the various assays and the feasibility of doing these assays in a one-step format are now being discussed throughout the results section as well as in the introduction and discussion.

The term 'therapeutic window' is generally understood to mean the range of doses/concentrations that are therapeutic, without causing toxicity. The lower limit, or trough concentrations, for TNF inhibitors varies depending on indication (Aliment Pharmacol Ther. 2020 Mar;51(6):612-628.). From reference 51 of the manuscript, the trough concentrations for infliximab are 3-7 µg/mL, which would be ~20-47 nM. Even these trough values are above the grey-boxed area labeled as "therapeutic window" in Figure 4E. As such, the assay appears to be more calibrated with sub-therapeutic concentrations than to clinically relevant levels.

The reviewer is correct, the gray-boxed area indeed indicates the typical cut-off values and trough values for these antibodies (i.e. the concentrations just before a new dose is administered). The concentration values on the X-axis represent the concentration in the final assay, i.e. after 10-fold dilution of the plasma samples. Therefore, the concentration window of 0.85-5.3 nM shown for adalimumab corresponds to 8.5-53 nM concentrations in undiluted plasma. We have adjusted the figure and the text to make clear that these values refer to cut-off values and trough values that are therapeutically relevant and not the classical definition of therapeutic window. While we focus on cut-off values and trough values here, measuring higher antibody concentrations is of course also possible by further dilution of the sample. We updated the text in this section and added the relevant references. ("Because of the relatively high cut-off values and trough concentrations of adalimumab and infliximab⁴⁵⁻⁴⁸, assays were performed using higher sensor concentrations (10 nM anti-antibody-LB and 100 nM TNFα-SB)." and "Gray boxes represent the clinically relevant concentration range of each therapeutic antibody from cut-off value to trough concentration (8.5–53 nM and 3.3–47 nM for adalimumab and infliximab, respectively⁴⁵⁻⁴⁸), corrected for dilution.")

The majority of assays have a downward trend in luminescence or ratio at higher analyte concentrations. For example, (Figure 4G, Supplementary Figure 8A) signals increase to a max of ~ 10 nM, but decreases above that. Is this a result of substrate depletion? What amount/concentration of NanoLuc substrate was used? Although not indicated specifically, results appear to be from an 'end-point' read of luminescence values, instead of a kinetic evaluation (eg initial velocity). Would using the latter allow for higher concentrations of analyte to be assessed?

The downward trend at higher analyte concentration is not a result of substrate depletion, as it is also observed using ratiometric detection with a calibrator luciferase. This trend results from what is known as the 'hook effect' and occurs because at high target concentrations the two sensor components no longer bind simultaneously at the same analyte but are instead 'diluted' over the many analytes present. This results in a bell-shaped curve at analyte concentrations that exceed the sensor concentrations (also depending on antibody affinity), which is also predicted by the thermodynamic model. Although we already discussed this in the original manuscript, we have adjusted this explanation to make it more clear. ("The luminescent output follows a bell-shaped dose-response curve, as evidenced by a decrease in signal at cTnI concentrations >10 nM (Fig. 2b). This 'hook' effect is well-known and occurs because at high target concentrations the two sensor

components no longer bind to the same target to form an enzymatically active ternary complex, but instead each bind to different target molecules^{31,32}.”) The additional kinetics data that we now provide also make it more clear that substrate depletion can result in a decrease in absolute intensity, but does not affect the emission ratio. To allow the quantification of higher concentrations of analyte (apart from dilution) one could increase the concentration of the sensor components, preferably the component that contains Gx-SB, as this small luciferase fragment does not contribute to background luminescence. This strategy was applied for the adalimumab assays (Figure 4E), which were done with 100 nM of the TNF α -SB fusion protein, ensuring that up to 100 nM analyte all antibodies were TNF α -bound.

The authors indicate that plasma concentration negatively impacts the luminescence (Supplementary Figure 11). Why was plasma used instead of serum? Does the fibrinogen present in plasma negatively affect the assay? In relation, the amount of plasma in each different assay is not clearly indicated. Was the dilution for each assay consistent? Including these parameters (substrate concentration/dilution, incubation time, run time, etc) would be a simple and valuable addition (perhaps added to the Supplementary Table).

One reason to use plasma is that this is slightly more relevant for POC assays, for example for those that use cell separation filters. In response to this question we have now compared the performance of our assays in plasma and serum (Supplementary Figure 14) and see a similar behavior, suggesting that the effect is not related to the presence of fibrinogen in plasma. We updated the text accordingly. (“Additionally, we assessed sensor performance in human blood plasma and serum. The luminescent output signal was substantially suppressed in 100% plasma, which can be partially attributed to absorption of blue light by hemoglobin and biliverdin (Fig. 2h and Supplementary Fig. 13)^{33,34}. Nevertheless, the high intrinsic sensitivity of the RAPPID sensor still enabled ratiometric detection, with a reliable >5-fold change in emission ratio in both 20% blood plasma and serum (Fig. 2h and Supplementary Fig. 14).”) The dilution used in the various assays was not only guided by the effect of plasma/serum on the assay response (this only requires a 5-fold dilution), but also guided by the physiologically relevant concentrations. For example, a 1000–4000-fold dilution was employed for the quantification of CRP in patient plasma samples. As requested by the reviewer we have indicated the amount of plasma used for each assay, as well as all other relevant assay parameters (Supplementary Table 1).

In assays presented in Figure 4 C, authors state that “Type 2 or 3 anti-antibody” are used. Which type was used for each of the assays?

A type 3 antibody was used for the adalimumab assay, while a type 2 antibody was used for the infliximab assay (due to a lack of type 3 antibody). We have added this information in the main text. (“To overcome this issue, we implemented an alternative RAPPID format by introducing a direct genetic fusion of SB to TNF α , and combined it with a Gx-LB-photoconjugated type-2 anti-antibody, binding to the therapeutic antibody infliximab, or a type-3 anti-antibody, binding to the complex of the target therapeutic antibody adalimumab and TNF α (Fig. 4c and Supplementary Fig. 24)”) As the type 3 antibody specifically recognizes the antibody-TNF α complex, this assay does not display the Hook-effect at higher antibody concentrations.

How does the selection of these respective antibodies change the kinetics of the assay, and therefore affect the optimal read-time?

We have now studied the kinetics of the formation of the sandwich complex for both adalimumab and infliximab and found that their kinetics are quite similar, despite the use of different types of detection antibodies. As expected the kinetics of complex formation is strongly dependent on the concentration of the target antibody, but the rate of complex formation can also be enhanced by increasing the concentration of either TNF α -SB (see new data in Figure 4F) or by increasing the concentration of the Ab-LB (see Supplementary Fig. 25 and 26). (“Kinetic experiments revealed that increasing the concentration of TNF α -SB markedly decreased the time to reach stable sensor output, from >30 min to less than 15 min for detecting 1 nM adalimumab or 1 nM infliximab (compare columns in Fig. 4f). At the high sensor concentrations used for the assays described in Fig. 4e (10 nM antibody-LB, 100 nM TNF α -SB), formation of the sandwich complex is complete even within 5 min (Supplementary Fig. 25b and 26b).”)

Also, the intrinsic activity of Large BiT, as well as auto-complementation appear to be major limiting factors. Do the authors have insight into potential solutions, or is the background luminescence unavoidable?

First, it is important to note that the background is relatively low (e.g. $\sim 0.3\%$ of that of the complemented activity for the cTnI assay, Figure 2B). At sensor concentrations of 1 nM, the background activity is mainly due to the intrinsic activity of LB itself, as evidenced by the experiments shown in Supplementary Fig. 8A. At higher sensor concentrations, both experiments and modeling show that some of the background activity is also due to complex formation in the absence of analyte. This background complex formation could be suppressed by attenuating the affinity of the small bit component (Supplementary Fig. 9B), but this is detrimental to complementation in the sandwich complex. While we had included these data in the supporting information, we now discuss them more extensively and explicitly in the main text (*“Control experiment show that the low background activity that is observed under these conditions in the absence of analyte (<0.5%) is primarily caused by residual activity of uncomplemented LB and to a lesser extent by the intrinsic affinity between split fragments LB and SB (Supplementary Fig. 8a).”*). Very recently, Hall and coworkers also reported the development of a triple split system, where the background activity of the large Bit component was completely abolished (ref 25). However, full complementation for this system also required higher concentration of sensor components, which potentially can result in higher background complementation. Nonetheless, combining this system with the RAPPID platform is certainly worth exploring.

When selecting a pair of antibodies that bind a single antigen, does the distance between respective binding epitopes of the antibodies make a difference in signal? Are there limitations to distance? For configurations that require a single antibody to simultaneously bind two different target antigens (direct antibody monitoring, Figure 4F), what are the limitations in size or orientation of the analyte?

While the distances between respective epitopes on the antigen (and the binding orientation) can be expected to affect the efficiency of complementation (by its effect on the local effective concentration of the split luciferase components), we have not yet encountered an example where the RAPPID assay did not work when using antibodies that recognize non-overlapping epitopes. The semi-flexible linkers that we employed are based on the linkers used in our LUMABS sensors and are known to effectively bridge the 10-15 nm distances encountered in sandwich immunoassays. Nonetheless, further optimization of linker properties (length, but also stiffness) may result in more efficient complementation for a specific antigen-antibody pair combination. We have added the following sentence to the discussion to provide more insight into this issue (*“The semi-flexible linkers that were employed are based on previous luminescent sensors and known to effectively bridge the 10–15-nm distances encountered in sandwich immunoassays, and RAPPID assays were successfully developed for all antigens tested so far, representing a wide variety of architectures. Nonetheless, depending on the architecture of the sandwich complex, optimization of linker length and stiffness could be employed to further improve complementation efficiency for specific targets.”*). Similar considerations apply to the assay format for direct antibody monitoring using split components genetically fused to the antigen (Figure 4F in the original manuscript, now Figure 5D,E). The linkers that are used can effectively cover the 10–15 nm distance between the two antigen binding domains in the target antibody. This is expected to work for most antigens, unless (1) the antigen domain itself is very large (>5 nm) and (2) the linker is fused at a location in the antigen that in the antigen-antibody complex is pointing away from the other binding domains. In the latter case longer and stiffer linkers could be employed or the linker could be introduced at a different position in the antigen.

Protein G conjugation of small BiT to antibodies does not seem to go to completion. Do the authors have any insight into why this is the case (Supplementary Figure 17,18) and how it could influence variability in the read-out of the assays?

Our experiences with photoconjugation are in line with those reported in the original LASIC paper in terms of specificity (ref. 26), but we do indeed observe some differences in the efficiency of conjugation for different antibody types. The reason for these differences are not clear but may result from small variations in heavy chain subtypes, glycosylation, etc. We typically observe that the efficiency of photoconjugation can be pushed to completion by increasing the Gx-LB/SB-to-antibody ratio, but the increase of non-conjugated Gx-LB and Gx-SB might also result in an increase in background signal. As described, we therefore typically chose to use 1:1 to 1:4 molar ratios of antibody to Gx-LB/SB protein to minimize the presence of non-conjugated Gx-LB and Gx-SB, yielding a mixture of once- and twice-conjugated antibodies. We did not encounter significant differences in assay performance between batches with different conjugation efficiencies, provided that the calibration is done with the same batch. We are currently exploring protein engineering approaches to further enhance conjugation efficiencies that also yield once conjugated antibodies exclusively. (*“Although the conjugation efficiency can be further improved by increasing the relative concentration of Gx-LB and Gx-SB, we chose to*

use an equal molar ratio of protein to antibody to minimize the presence of non-conjugated Gx-LB and Gx-SB, which could contribute to non-specific background complementation".)

It is surprising, and unacceptable, that there is no statistical analysis of any of the results. The only exception is Figure 3C, indicating Pearson's correlation.

We are not completely sure what kind of statistical analysis the reviewer is referring to. All experiments have been done at least in triplicate and are shown as individual data points or as mean values with error bars indicating s.d.. LOD values are reported with a 95% confidence interval and listed in Supplementary Table 1. Pearson's correlation values were determined when comparing the results of two different methods (comparison of plate reader and digital camera detection (Figure 2G); comparison of CRP detection by ELISA and RAPPID (Figure 3D); comparison of CRP detection in individual patient samples between RAPPID and clinical assay (Figure 3F)). For the clinical validation study in individual patient samples, we performed Bland-Altman analysis to assess potential bias (Figure 3G). Finally, we have now calculated values for Z' for each of the assays reported, as also requested by reviewer 2 (Supplementary Table 1).

Reviewer 2

In the present manuscript, Ni et al described a novel immunoassay approach to detect a variety of analytes by coupling antibodies with a luciferase complementation method. This is a very interesting manuscript and well-presented manuscript and while this has similarities with recently described work see (Hwang et al, 2020) the innovation of this approach is in coupling with an internal luciferase 'calibrator' to improve the relative stability of the assay through ratiometric detection and as the authors demonstrate the relative adaptability of this approach to a number analytes in multiple formats. I will be of interest to see which of the potential benefits and future uses and of these approaches discussed by the authors come to fruition.

We thank the reviewer for her/his positive feedback and appreciation of the importance of the ratiometric detection and modularity. As a first demonstration of the potential benefits we have now included a clinical validation of the RAPPID assay for CRP that directly compares the performance of the assay with a CRP assay currently used in clinical laboratories (Figure 3).

I have a number of additional comments for the authors to consider.

Comments:

While noted in the results (ln 109) it would be useful for the authors to comment of the potential effect of bi-conjugation of the NanoBiT fragments to the antibodies and if this would affect the results. Particularly which respect to the LgBiT fragment which can have detectable luminescence without complementation.

We already partially addressed this issue in response to some questions of reviewer 1. LB indeed displays some (but very low) residual activity, which is mainly responsible for the background signal in the absence of analyte. In this respect antibodies containing two copies of LB could result in a slightly higher background activity (max. two-fold) compared to an antibody with a single LB domain. However, it is also possible that in some sandwich complexes both LB domains are complemented, resulting in a higher signal in the presence of analyte. Quantifying the importance of this would require careful controls with antibodies conjugated with either once or twice to LB or SB domains and may also differ between analytes. However, since the effect can maximally increase the background by a factor 2 and we typically observe 20-200 fold increases in luminescence activity, this does not represent an important practical limitation.

The authors state that that the data represent triplicate experiments. It would be useful to know if this was three independent experiments or the one performed in triplicate.

The data represent a single experiment performed in triplicate, wherein the analytes were independently prepared by dilution. We have added this information to all the figure captions.

Furthermore, there appears to be significant variation in some assays (see Figure 2 C/D), which appears random between doses. Indeed, a key conclusion by the authors is that the "ratiometric signal is stable over time and less susceptible to experimental variation" and that the future use of these assays as point-of-care diagnostics

that would need to be highly reproducible between samples. An analysis of the reproducibility eg (Z' factor or similar) of one the assay formats would support such claims of utility.

The variation in Figure 2C,D indeed results from the intrinsic variation associated with intensimetric bioluminescent assays, in which small differences in incubation time, substrate concentration, and temperature between replicates can give rise to different enzyme activities and thus signal intensities. In fact, this is an important argument for ratiometric signal detection as introduced here with the calibrator luciferase, where visual inspection indeed shows much lower variation in all ratiometric assays reported here. To further support this claim more quantitatively we calculated the Z' factor for all RAPPID assays reported in the manuscript, as suggested by the reviewer. Z' factor values of 0.9 are typically observed, with the poorest assay still showing a very reasonable Z' of 0.71. The results of Z' factor analysis have been added to Supplementary Table 1) and are now also discussed briefly in the main text. (“Although the dynamic range is lower than in the intensimetric assay due to spectral overlap between blue and green emission, the >25-fold change in emission ratio is robust (Z' = 0.95, see Supplementary Table 1) and greater than those typically seen in BRET-based sensors^{10,15”}).

On page 6 the authors benchmark the RAPPID assays with current ELISA approaches and report similar/more sensitive detection with the RAPPID approach (pM-nM) v (8-80nM) however the data presented in supplementary Figure 13 appear to show that that RAPPID assays is 1000-fold less sensitive than ELISA (mg/L v ug/L). Is there are technical reason for this?

In this particular case the RAPPID assay is sensitive in the range of 10 pM–2 nM (Figure 3), which corresponds to 1–250 µg/L (Supplementary Fig. 19). The ELISA used here was more sensitive than the RAPPID assay and needed to be diluted even further, yielding a sensitivity range between 0.1 µg/L and 1 µg/L. The sensitivity of both assays is more than sufficient to detect the 1–10 mg/L sensitivity (8–80 nM) required for cardiovascular risk assessment. The most common application of CRP measurements is to detect and distinguish between bacterial and viral infections, in which case CRP concentrations are another 2 levels of magnitude higher (2–200 mg/L). This is the application for which we performed the clinical validation in Figure 3F. To this end, plasma samples were diluted 1000-fold or 4000-fold in the RAPPID assay to get in the responsive range of the sensor. We have included an extensive explanation about the clinical validation of the CRP RAPPID sensor in the legend of Supplementary Fig. 20.

In addition, it is not clear why the RAPPID approach is appears to be detecting almost twice as much CRP compared to ELISA in Figure 3C presumably from the same starting concentrations? Clarification of this would be useful.

Indeed, we observe good, linear correlations between the RAPPID assay and the ELISA (Figure 3d), but also for patient serum samples between the RAPPID assay and the standard automated immunoturbidometric assay used in the clinic (Figure 3f). In fact, the RAPPID assay and the clinically used automated immunoturbidometric assay yield very similar CRP concentrations, with only a small ~10% deviation of the theoretically expected slope of 1. The reason for the 1.5-fold difference between RAPPID and the ELISA is presently unclear, but maybe related to the multivalent nature of CRP. It is not uncommon for different ELISAs to report slightly different absolute values of analyte (see for example Schmitz et al. 2016, PMID: 26587745). The reasons for this are not always clear, but can be corrected for between different assays as long as there is a clear linear correlation.

Throughout the manuscript the Ni et al use SB/SmBIT to indicate the small fragment of Nluc (VTGYRLFEEKES) with an affinity of K_d 2.5 uM (peptide 101, Dixon et al 2016). Current nomenclature from the manufacture and that which prevails in the literature is that peptide 114 (VTGYRLFEEIL, K_d 190 uM, Dixon et al 2016) is designated SmBIT. Standardisation of this in the manuscript would assist the reader. In addition, for clarification in SuppFig22 sequences should be labelled SB2 and SB3 as per SuppFig9

The manufacturer indeed uses the terms SmBit and HiBit to refer to the small fragment variants binding with low (K_d of 190 µM) and very high affinity (K_d of 0.7 nM). To prevent misunderstanding we have decided to consistently use LB and SB to refer to the large protein component and small peptide component of the assay, respectively. LB actually corresponds to LargeBit, whereas we now explicitly state the peptide variant we refer to with SB is the peptide variant with a K_d of 2.5 µM. This variant is indeed different from the SmBit variant often used to detect protein-protein interactions, which has a K_d of 190 µM. We added a sentence in the first

paragraph of the results to clarify it. (“A variant of Small BiT with $K_d = 2.5 \mu\text{M}$ was chosen to ensure effective complementation in the sandwich complex (vide infra)²³.”). We have also labeled the sequences in Supplementary Fig. 35 (previous Supplementary Fig. 22) as SB2 and SB3, as suggested by the reviewer.

Minor comments:

In Figure 2b is insert up to 100pM or 10pM?

It is up to 10 pM. The position of the dash-line rectangle is corrected accordingly. We apologize for the mistake.

There is no mention of Supplementary Figure 9 in the manuscript and would be useful to have inserted on page 4 lines 165-166 to support the statement.

We agree and apologize for the omission. Reference to Supplementary Fig. 9 has been added to the text. (“Experimentally, increasing the strength of the interaction ($K_d = 0.28 \mu\text{M}$) does not enhance luciferase complementation much further in the presence of the analyte, but does results in a higher background signal in the absence of analyte and therefore a smaller dynamic range (Supplementary Fig. 9).”)

Can more detail as to how the LOD was calculated be provided in the methods, and as above whether this was calculated from a single experiment performed in triplicate or from three separate experiments. Notably, the LOD seems to be variable between the cTnl assays Fig2b (LOD 4pM) and SuppFig8a where there does not appear to be a detectable signal at 5pM?

The LOD was calculated from a single experiment performed in triplicate (wherein the analytes were independently prepared by dilution) by linear regression of the response related to the analyte concentration for a limited range of concentrations. An explanation of the calculation is given in Supplementary Table 1. Indeed, the LOD slightly varies between assays, especially for intensimetric assays in which small differences in substrate concentration, incubation time and temperature between replicates can give rise to different signal intensities. The LOD of the intensimetric cTnl assays shown in Figure 2b was calculated to be 4 pM, while that of the intensimetric assays shown in Supplementary Fig. 8B was calculated to be 18 pM.

Reviewer 3

In the presented manuscript Merx and colleagues continue their long-term search for technologies that would allow rapid development of protein biomarker quantification assays. In this study they depart from their earlier efforts to build fully integrated bespoke biosensors systems and, instead, develop a molecular tool kit and its application protocols that allows rapid immune assay development for a broad range of protein analytes.

A person familiar with the earlier research outputs of the group can see the evolution of ideas that led to the current development. In the present study the authors apply the idea of ratiometric bioluminescence sensing, that was originally developed for BRET sensors, to split enzyme systems thereby getting around the problem of time resolved changes in emission that complicates measurements using Luciferase -based systems. They also borrow from the field of chemical biology in order to rapidly produce covalently linked antibody- reporter conjugates through crosslinking of genetically incorporated photoactivatable amino acid. They analyse in detail the optimal ways to minimise the “hook effect” – an important issue that arises from the use of non-cooperative target binders. They test the developed assay on a range of targets and provide data on its performance when monitoring the assay using simple consumer camera.

From the technical perspective the paper combines, improves, and optimises a range of pre-existing technologies to develop a new assay format. While these are interesting and well executed the novelty is not very high and this shifts the attention to the potential practical utility and impact of the developed platform.

We thank the reviewer for the detailed analysis of the different aspects of the work presented in our manuscript. The strength of the work is not only in the clever combination of pre-existing technologies in a novel assay format. The concept of using a calibrator luciferase to render luciferase-based assays ratiometric is completely new and has important advantages and broad practical applicability, as also recognized by the other reviewers.

The authors state that the platform “provides an attractive, fast, and low-cost alternative to traditional immunoassays, both in an academic setting, in clinical laboratories and for point-of-care applications.”

In my opinion an improvement to the cost structure for the assays performed in academic laboratories, constitutes a modest advancement and does not need to be evaluated in detail.

I do not think the authors made a convincing case that the presented technology has the potential to either replace or augment existing commercial solutions used by diagnostic laboratories. While the contemporary chemical pathology is using a lot of legacy technologies, the industry had decades and billions of dollars to optimise their use and to achieve very impressive accuracy and efficiency. For example a good comparison case is the CEDIA system (Thermo-Fisher) that is based on split beta-galactosidase. Given that the technology stems from 1980th it is using less elegant solutions than that in the present study but the fact that it is a clinically used assay platform suggests that the robust performance has been achieved.

Potentially, one can argue that the developed technology allows hardware downgrade by replacing complex immunochemistry stations with less sophisticated devices. However, unlike absorbance-based chemistry analysers, homogeneous luminescence assays are lot less frequently used in diagnostic practice and hence the existing hardware platforms cannot be used.

While our assay technology has indeed fundamental advantages compared to classical, absorbance-based heterogeneous immunoassays currently used in central clinical chemistry laboratories, we agree that the amount of money and time that was invested in their optimization, automation and commercialization will present a commercial barrier to replace these highly automated processed in the near future. As such, we agree that the most immediate applications are envisioned in research laboratories and in POC settings. However, we do believe that the intrinsic advantages connected with homogeneous bioluminescent assays also renders them attractive for use in clinical chemistry laboratories, although initially primarily to augment existing assays or in clinical laboratories with limited resources. The argument that at present homogeneous luminescence assays are used a lot less frequently is a circular argument, since these types of assays are only being developed now and the RAPPID platform actually represents the first generic sandwich immunoassay platform that can be broadly applied. Bioluminescent homogeneous detection indeed allows for a substantial downgrade in hardware requirements, as we demonstrated in the validation of the RAPPID assay for CRP on 40 different patient samples in the clinical chemistry laboratory of the Rijnstate hospital. The hospital did not have equipment for multicolor luminescent detection, therefore we simply used a normal digital camera for assay readout, yielding quantitative results that correlate very well with CRP values obtained on the highly automated random access analyzers used in their clinical chemical laboratory.

While, based on the evidence provided by the authors , their replacement may be technically and commercially possible it would require regulatory approval of a new class of medical devices which is cumbersome and is likely to have limited appeal to their users. I think a well-argued case needs to be made to support the utility of the developed technology for the laboratory testing (see below) and at least the back of the envelope calculations would be needed to support authors assertion.

The need for regulatory approval is true for any new class of medical device and can used as an argument against any new development in diagnostics. As discussed more extensively below, we have now assessed the performance of our CRP RAPPID assay on actual clinical samples and show that its analytical performance is comparable to the highly automated method currently used in the Rijnstate hospital clinical chemistry laboratory. Again, we realize that the first commercial applications will not be to replace the highly automated immunoassay platforms, but there remain plenty of applications where such equipment is simply not available, such as in a general practitioner's office. We also do not claim in our manuscript that we envision the RAPPID assay to easily replace clinical laboratory testing, and instead emphasize that we believe its main potential is in point-of-care diagnostics. ("Such technology would not only accelerate biomolecular research in academic and clinical environments, but more importantly, allow for fast diagnostic decision making when access to clinical laboratories is unavailable or cost-prohibitive." and "Single-step affinity-based detection assays performed directly in solution could not only speed up traditional laboratory immunoassays, but would be particularly attractive for point-of-care testing outside of the laboratory setting by non-expert users^{4,5}." and "In conclusion, RAPPID represents an attractive new immunoassay format for point-of-care diagnostics, that combines the analytical performance of a laboratory-based immunoassay such as ELISA with the speed and ease-of-use of lateral flow immunoassays.").

Point-of-Care. As COVID-19 pandemic convincingly showed development of rapid and cheap distributed biochemical diagnostics is yet an unsolved problem. At present there is no clear technological leader there as all, so far, reported approaches have their limitations. The authors are the leaders in this field and therefore

should be able to comprehensively and critically assess the potential of the developed technology for PoC applications.

In the view of the above here are the issues that the authors should try to address in the revised version of the manuscript:

1) Assay format. At present the assay is composed of a range of reagents that also includes NanoGlo (furimazine) substrate from Promega of an unknown concentration. How stable is this substrate and in what format would it be utilised in the PoC assays?

In our work on paper-based analytical devices (ref. 18) the stability of the furimazine substrate towards auto-oxidation was indeed found to be bottleneck for long-term storage at room temperature over a period of days or weeks. The oxidation sensitivity of the substrate could be addressed by several technological solutions such as packaging under low-oxygen conditions. Very recently, the Promega team reported a formulation that apparently allows drying in of the furimazine substrate and split Nanoluc that is stable for months at ambient conditions. We now discuss this important issue for POC applications and refer to this work that was published last month. (“A possible limitation of the current RAPPID assays for POC applications, is the limited long-term stability of sensor components at room temperature, in particular for the NanoLuc substrate. However, this could be addressed by packaging devices under low-oxygen conditions, or by applying a recently reported formulation of an all-in-one reagent that has a long shelf life at ambient temperatures²⁵.”).

2) Pre-incubation times and the total duration of the assay. These are very important parameters that define technical implementation of the test and its clinical utility as, for an example, in many countries doctor visits are short (10-15 min) and the data obtained outside of this time window is less competitive with the established diagnostic methods. Further, longer measurement times indirectly influence the accuracy of the test as at prolonged incubation leads to variability due to sample drying, temperature fluctuations, user-inflicted environmental changes etc. Furthermore, the requirement for two step assay (Step1: sample+ antibody conjugates, Step 2: NanoLuc substrate addition and emission measurement) also significantly complicates design of the test's hardware. The Results and M&M sections refer to pre-incubation times of sample with the protein components of 5-60 minutes. The authors need to provide a time dependent activation data for at least couple of assays for unrelated analytes where they mix the sample and then withdraw aliquots at different incubation times followed by addition of the NanoGlo and data collection. The analysis of the dynamic ranges and CV at different incubation times should be instructive in estimating of the minimal time this assay can be performed at. Finally, one would expect that antibodies would bind to their targets in seconds if not faster. If extended incubations are required than what step of the process do they support and what can be done to shorten this time

This point was also raised by reviewer 1. As requested we have now obtained time-dependent measurements for several different analytes in which we monitored formation of the sandwich complex in real time immediately after adding the analyte (Supplementary Fig. 12, 16, 22, 25, 26 and 32). Instead of taking aliquots at different incubation times, the ratiometric nature of the assay allowed us to monitor the reaction directly over time. As discussed in response to a similar question by reviewer 1, the kinetics of complex formation varies between assays and depends on the analyte concentration, the concentration of the sensor components and the specific antibodies that are used. In some assays such as the Adalimumab and Infliximab assays, equilibration is reached within 10 minutes (Fig. 4f and Supplementary Figure 25 and 26). In others, full equilibration requires longer times, but also here reliable quantification is often feasible after 5–10 minutes, provided incubation times are controlled.

3) Assay temperature sensitivity. How sensitive the assay to the temperature fluctuations? This is a critical parameter in PoC testing and while it can be controlled it comes at increasing cost and complicates technical implementation. This may be related to the point 2.

This is an important point, as temperature indeed is an important parameter for any enzyme-based assay. In case of luciferase-based assays temperature both affects the activity of the luciferase and the rate of substrate depletion, both of which can be addressed by using a calibrator luciferase and ratiometric detection. We have performed two experiments to compare the temperature sensitivity of intensimetric versus ratiometric detection. First, we compared the temperature sensitivity of Nanoluc activity (intensimetric signal) with that of a 1:1 mixture of Nanoluc and calibrator luciferase (Supplementary Fig. 17). As shown, the ratiometric signal remains constant between 20 and 35°C, whereas the intensimetric signal at 458 nm increases by 44% when

raising the temperature from 20 to 35°C. In a second experiment we compared the temperature sensitivity of the intensimetric signal (absolute intensity at 458 nm) and the ratiometric signal in a RAPPID assay for CRP. Again, a more variable sensor output was observed for the intensimetric assay than for the ratiometric one (Supplementary Fig. 18). These new experiments clearly show that ratiometric detection effectively addresses the inherent temperature sensitivity of luciferase-based assays. However, in specific cases temperature sensitivity could still arise from the temperature dependence of the antibody-antigen interaction (but this is true for any immunoassay). These new data have been included in the supporting information and are briefly discussed in the main manuscript. (“We also used the CRP assay to assess the effect of temperature on assay performance, which is an important feature to consider when testing at the point of care. While the absolute intensity of both the sensor and the calibrator luciferase increased by 30–40% between 20°C and 37°C, the ratio of the two signals, and thus the RAPPID sensor output, remained stable in this temperature window (Supplementary Fig. 17 and 18).”)

4) *Assay stability and comparative performance. The authors argue that the assay can be used to measure clinical biomarkers in human samples albeit with significant sample dilution. However, they do not provide any data on performance of the assays in the actual clinical samples and their benchmarking against currently used diagnostic methods. For instance, the data they provide on cardiac troponin is very interesting as reliable detection of cardiac troponin at lower concentrations is an unsolved problem in diagnostic industry (For instance : Outliers as a cause of false cardiac troponin results: investigating the robustness of 4 contemporary assays. Clin Chem . 2011 May;57(5):710-8. doi: 10.1373/clinchem.2010.159830 and Improved Diagnostic Performance of High-Sensitivity Cardiac Troponin Assays Is an Artifact of Censored Data. Clin Chem, 2016, 62(12):1654-1657. doi: 10.1373/clinchem.2016.262634). Benchmarking this assay against currently used diagnostic methods using real clinical samples even in solution format would significantly strengthen author’s case and also test the assay’s robustness against patient-originated experimental noise.*

As suggested by the reviewer we have now assessed the performance of one of our RAPPID assays on actual clinical samples and compared it against a method currently in use in the clinic. In collaboration with the clinical chemistry laboratory of Rijnstate hospital we used our RAPPID CRP assay to determine the CRP concentration in 40 individual patient serum samples and compared the values with those obtained by the automated immunoturbidometric assay currently used in this laboratory. An excellent correlation (Pearson’s $r = 0.993$, slope 1.08 ± 0.02) was observed between the RAPPID assay and the clinical method. We have included this important new data in Figure 3, and have added a detailed description of the experiments and a more extensive analysis of the data in the supplementary information (Supplementary Fig. 20). (“Further validation was performed by comparing RAPPID to an automated immunoturbidometric assay used in the clinic for routine measurements of CRP as an inflammation marker in blood plasma at high mg L^{-1} levels (approximately 0.03–2.4 μM). Dilutions of individual, freshly collected 1- μL patient plasma samples (total $n = 40$) were measured using RAPPID and detected with a digital camera. In parallel, samples from the same patients were measured using the clinical assay (Fig. 3e and Supplementary Fig. 20). Our results showed excellent linear correlation and a small proportional difference with the clinical assay (Pearson’s $r = 0.993$, slope 1.08 ± 0.02), and good reproducibility with an average coefficient of variation of $8\% \pm 6\%$ derived from technical triplicates (Fig. 3f,g). Combined, these validation studies illustrate that RAPPID enables accurate determination of biomarkers in a clinical setting.”)

5) *Can the assay be dried down and combined with blood filtering solutions that the authors used in their 2018 Angewandte Chemie paper? This would bring the system lot closer to a PoC format. The discussion of these aspects would be important.*

This is indeed an interesting question for follow-up research. As already discussed above, the Promega team very recently reported a formulation that apparently allows drying in of the furimazine substrate and split-Nanoluc that is stable for months at ambient conditions. In follow-up research we will certainly test whether these formulations can be combined with paper or thread-based microanalytic devices and the RAPPID assay. We now discuss these possibilities in the general outlook at the end of the manuscript. (“Ratiometric detection is also an important prerequisite for the successful application of bioluminescent assays in point-of-care diagnostics. Direct comparison of RAPPID with a commercially available ELISA kit and an automated clinical assay demonstrated that the blue-to-green ratiometric sensor signal can be detected fast and robustly with minimal equipment and low sample volumes, while mirroring the clinically approved assays in terms of

accuracy and sensitivity. Additionally, implementation at the point of care can be envisioned by integration of the bioluminescent sensor proteins into paper- or thread-based analytical devices^{11,19}. To increase the sensitivity of such devices, RAPPID could also be combined with dedicated microfluidic devices, while dedicated readers could be developed that capture the bioluminescent light more efficiently than commercial digital or smartphone cameras⁶³. A possible limitation of the current RAPPID assays for POC applications, is the limited long-term stability of sensor components at room temperature, in particular for the NanoLuc substrate. However, this could be addressed by packaging devices under low-oxygen conditions, or by applying a recently reported formulation of an all-in-one reagent that has a long shelf life at ambient temperatures²⁵.“).

6) Assay inhibition by serum. Authors show that plasma has inhibitory effect on the assay and make suggestions of the causes. This is a very important aspect as dilution both complicates the assay format particular for PoC applications and also reduces the actual assay's sensitivity. Understanding the origin of the inhibition would be important for assay optimisation. The authors conjecture that absorption of the blue emission may be one of the reasons for inhibition. Is this the primary reason? If so the reported variants of NanoLuc with long wavelength emission would be able to solve the problem.

This issue was already partially addressed in response to a question by reviewer 1. As mentioned, we have done additional experiments that show that the attenuation is not due to differences between serum and plasma (Supplementary Fig. 14). Absorption of blue light by bilirubin and related components could certainly be one of the reasons, as also observed previously by Johnsson and coworkers (refs. 10, 11). For this reason, developing RAPPID variants with different colors would indeed be an interesting approach that we aim to explore in the near future, in conjunction with developing more red-shifted variants of the calibrator luciferase. An additional hypothesis is that certain plasma components partially compete with NanoLuc complementation at high plasma concentrations, which is not easy to prove but worth looking into in more detail in future studies. However, in most cases the issue can be effectively addressed by 5-fold dilution of plasma/serum samples, due to the high inherent sensitivity of the RAPPID assay.

Reviewers' Comments:

Reviewer #1:

Remarks to the Author:

I appreciate the authors' full and respectful consideration of my feedback. Their responses, both the new experimental data and the changes to the manuscript, are detailed and address all of my concerns. The further validation and additional methodology substantiates their approach and facilitates the adaptation their assay.

Reviewer #2:

Remarks to the Author:

I thank the authors for a thorough revision of the manuscript and addressing my concerns. I have no further comments.

Reviewer #3:

Remarks to the Author:

The authors have addressed the concerns raised by me and other referees and complemented the study with important benchmarking data and additional analysis that allows to evaluate the presented technology with greatest level of confidence. Also, I must agree with author's comments that the concept of internal reference is novel at least in the application to luciferase biosensors. I believe that the manuscript is now suitable for publication and congratulate the authors to an important and well executed study!

Response to the reviewers

We would like to thank all three reviewers for their in-depth review of the original manuscript and useful suggestions, which substantially improved the manuscript. We were very pleased that all three reviewers were completely satisfied with the revised manuscript and unanimously advised acceptance without any further changes (see below).

Reviewer #1 (Remarks to the Author):

I appreciate the authors' full and respectful consideration of my feedback. Their responses, both the new experimental data and the changes to the manuscript, are detailed and address all of my concerns. The further validation and additional methodology substantiates their approach and facilitates the adaptation their assay.

Reviewer #2 (Remarks to the Author):

I thank the authors for a thorough revision of the manuscript and addressing my concerns. I have no further comments.

Reviewer #3 (Remarks to the Author):

The authors have addressed the concerns raised by me and other referees and complemented the study with important benchmarking data and additional analysis that allows to evaluate the presented technology with greatest level of confidence. Also, I must agree with author's comments that the concept of internal reference is novel at least in the application to luciferase biosensors.

I believe that the manuscript is now suitable for publication and congratulate the authors to an important and well executed study!